https://doi.org/10.1038/s41467-022-28287-8　　**OPEN**

# SARS-CoV-2 genomes from Saudi Arabia implicate nucleocapsid mutations in host response and increased viral load

Tobias Mourier [1,20], Muhammad Shuaib [1,20], Sharif Hala[2,3,20], Sara Mfarrej [1,20], Fadwa Alofi[4], Raeece Naeem [1], Afrah Alsomali[5], David Jorgensen[6], Amit Kumar Subudhi [1], Fathia Ben Rached [1], Qingtian Guan[1], Rahul P. Salunke[1], Amanda Ooi[1], Luke Esau[1], Olga Douvropoulou[1], Raushan Nugmanova[1], Sadhasivam Perumal[1], Huoming Zhang [1], Issaac Rajan[1], Awad Al-Omari[7], Samer Salih[7], Abbas Shamsan[7], Abbas Al Mutair[7], Jumana Taha[8], Abdulaziz Alahmadi[9], Nashwa Khotani[10], Abdelrahman Alhamss[11], Ahmed Mahmoud [12], Khaled Alquthami[10], Abdullah Dageeg[13], Asim Khogeer[14], Anwar M. Hashem [15,16], Paula Moraga[17], Eric Volz [6], Naif Almontashiri[12,18] & Arnab Pain [1,19✉]

Monitoring SARS-CoV-2 spread and evolution through genome sequencing is essential in handling the COVID-19 pandemic. Here, we sequenced 892 SARS-CoV-2 genomes collected from patients in Saudi Arabia from March to August 2020. We show that two consecutive mutations (R203K/G204R) in the nucleocapsid (N) protein are associated with higher viral loads in COVID-19 patients. Our comparative biochemical analysis reveals that the mutant N protein displays enhanced viral RNA binding and differential interaction with key host proteins. We found increased interaction of GSK3A kinase simultaneously with hyperphosphorylation of the adjacent serine site (S206) in the mutant N protein. Furthermore, the host cell transcriptome analysis suggests that the mutant N protein produces dysregulated interferon response genes. Here, we provide crucial information in linking the R203K/G204R mutations in the N protein to modulations of host-virus interactions and underline the potential of the nucleocapsid protein as a drug target during infection.

---

A full list of author affiliations appears at the end of the paper.

The emergence of novel severe acute respiratory syndrome coronavirus 2 (SARS-CoV-2), which causes the respiratory coronavirus disease 2019 (COVID-19), resulted in a pandemic that has triggered an unparalleled public health emergency[1,2]. The global spread of SARS-CoV-2 depended fundamentally on human mobility patterns. This is highly pertinent to a country like the Kingdom of Saudi Arabia, which as of 22nd February 2021 had a total of 374,691 cases and 6457 deaths[3]. The kingdom frequently experiences major population movements, particularly religious mass gatherings. For instance, during Umrah and Hajj roughly 9.5 million pilgrims visit two Islamic sites in Makkah and Madinah annually[4,5] and the Ministry of Health takes public health measures to keep the pilgrims safe and major outbreaks have been by and large avoided in recent years. Further, an estimated 5 million Shiite Saudi nationals travel to Iran for pilgrimage, which became an early source of COVID-19 infections in the region[5,6]. This movement has been reflected in the early phase of COVID-19 transmission within Saudi, as the first case was officially reported in Qatif (Eastern Region) on March 2nd, 2020[7].

Genomic epidemiology of emerging viruses has proven to be a useful tool for outbreak investigation and tracking the pathogen's progress[8,9]. As of October 2021, over four and a half million complete and high coverage genomes are accessible on GISAID[10,11], which aids immensely in tracking the viral sequences globally[12]. Novel SARS-CoV-2 variants are continuously arising and besides providing signals for epidemiological tracking, a subset of the resulting variants will have a functional impact on transmission and infection[13–15]. It is therefore critical to monitor the genetic viral diversity throughout the pandemic.

In this study, we sequenced 892 SARS-CoV-2 genomes from nasopharyngeal swab samples of patients from the four main cities, Jeddah, Makkah, Madinah, and Riyadh, as well as a small number of patients from the Eastern region of Saudi Arabia (Fig. 1a, b, Supplementary Data 1, Supplementary Table S1, and Supplementary Table S2). Samples were mainly collected early in the pandemic and during the first wave of reported cases in Saudi Arabia (Fig. 1c). We analyzed the genomes to investigate the nucleotide changes and multiple mutation events that represent the first 6 months of the locally circulating pandemic lineages of the SARS-CoV-2 in Saudi Arabia and searched for the potential association of polymorphic sites in the genome with available hospital records including severe disease and case fatality rates among the COVID-19 patients. We performed phylogenetic analysis to visualize the genetic diversity of SARS-CoV-2 and the nature of transmission lineages during March-August, 2020. We have presented a snapshot of the genomic variation landscapes of the SARS-CoV-2 lineages in our study population and linked a specific set of mutation events in the N gene to viral loads in a diverse population of COVID-19 patients in Saudi Arabia (Supplementary Fig. S1). Finally, we experimentally show the functional impact of these mutations in the N protein on the virus' interactions with the host.

## Results

### SNP calling and phylodynamics of SARS-CoV-2 samples from Saudi Arabia.
We sequenced and assembled SARS-CoV-2 genomes from 892 patient samples. This group includes 144 patients that were placed in quarantine and had either mild symptoms or were asymptomatic. The remaining patients were all hospitalized (Supplementary Table S1). Data on comorbidities were available for 689 patients with diabetes (39%) and hypertension (35%) being the most abundant (Supplementary Table S2). Patient outcome data were available for 850 samples, and 199 patients (23%) died during hospitalization (Supplementary Table S1).

From the 892 assembled viral genomes collected over a period of 6 months, we found a total of 836 single-nucleotide polymorphisms (SNPs) compared to the SARS-CoV-2 Wuhan-Hu-1 isolate reference (GenBank accession: NC_045512) (Supplementary Fig. S2). The observed numbers of SNPs relative to the reference sequence are in general lower than the numbers observed in global samples, but with the exception of a period from mid-June to late July, the average number of SNPs in Saudi samples is within one standard deviation of samples deposited in GISAID (Supplementary Fig. S3). We further detected 41 indels of which 26 reside in coding regions (Supplementary Table S3). Most indels were specific to a single sample, and no identical indel was found in more than four samples. Compared with global SNP data, seven SNPs were found in higher frequencies (absolute difference > 0.1) in samples from Saudi Arabia (Supplementary Fig. S2). These include the Spike protein D614G (A23403G) and three consecutive SNPs (G28881A, G28882A, and G28883C) causing the R203K and G204R changes in the nucleocapsid protein. Together with all sequences from Saudi Arabia available on GISAID on December 31st 2020, the assembled sequences were used to construct the effective population size and growth rate estimates of SARS-CoV2 over the course of the first wave of the epidemic. The skygrowth model[16] (Fig. 1d) shows a downward trend in the effective reproduction number ($R(t)$) over time with the timely introduction and maintenance of effective non-pharmaceutical interventions by the Saudi Ministry of Health. Despite the bounds of $R(t)$ estimate including one for much of the study period, we can infer an estimated decrease in R(t) over time until the lifting of restrictions in late June (Fig. 1d). By investigating the 1.6 million individual traces from the Skygrowth analysis we can infer a date of 27th April 2020 as the first point where over 50% of collected traces resulted in an $R(t)$ estimate below one, suggesting an epidemic in decline for the first time.

The effective population size (Ne) represents the relative diversity of the sequences collected in Saudi Arabia over the course of the outbreak (Fig. 1d). The model predicts a peak in viral diversity at the beginning of June. This is ahead of the peak number of cases reported nationally and is likely influenced by the earlier peak in reported cases in the three cities, which contribute the most viral sequences to this analysis (Madinah, Makkah, and Jeddah).

A maximum-likelihood phylogenetic analysis revealed that samples from Saudi Arabia represent five major Nextstrain clades[12], 19A-B and 20A-C (Fig. 2a). The approximate relationships between Nextstrain clades and Pangolin lineages[17] are listed in Supplementary Table S4. The phylogeny highlighted the clade 20A that all carried the nucleocapsid (N) protein R203K/G204R mutations[18] with high incidences of ICU hospitalizations. These samples were predominantly coming from Jeddah. Through time-scaled phylogenies dates of importation events were then estimated for each clade. The estimates of importation date with phangorn (see "Methods") are early, with clade 19B estimated to have been imported with the highest probability in late February 2020 and the other reported clades most likely imported in March (Fig. 2b). As this method cannot directly estimate the timing of an importation event, importations were estimated to occur at the midpoint of a branch along which a state change occurs between the internal and external samples. As estimated branches are short due to the inclusion of closely related external sequences this should not significantly alter the estimated importation dates but does make an assumption about the timing of the movement event which could be overcome with more complex and

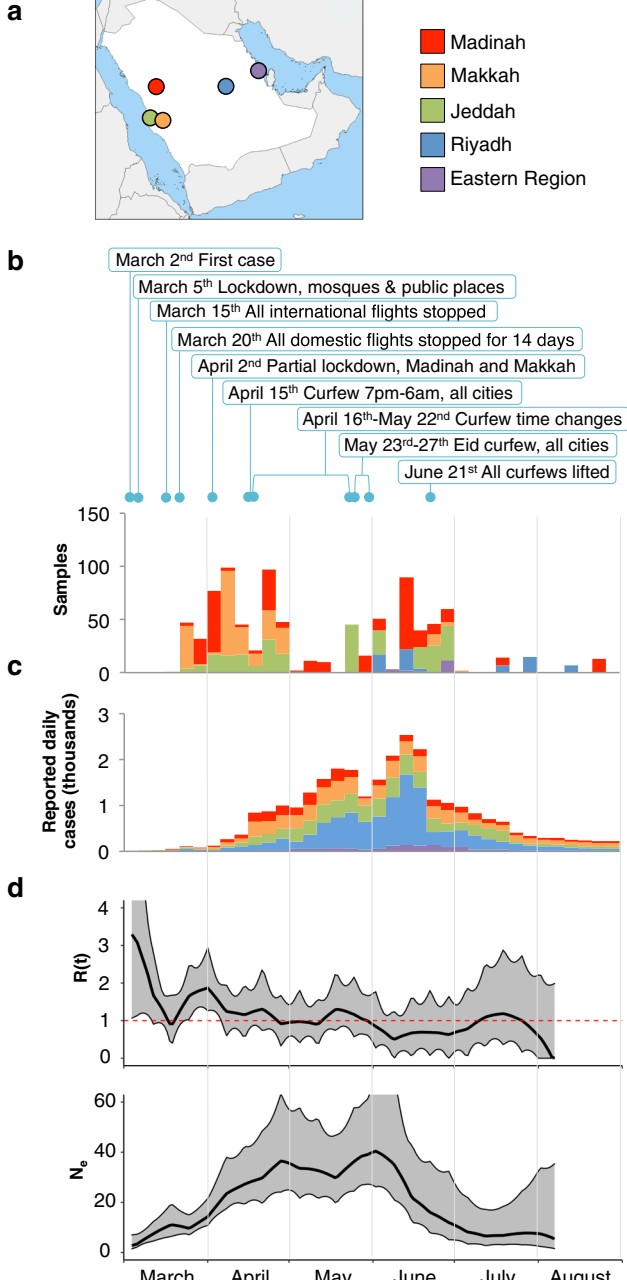

**Fig. 1 Sample overview and population genetics. a** Locations of the sampling cities within Saudi Arabia. **b** Stacked bars showing the numbers of samples retrieved from the 4 cities and the Eastern region during the first six months of the pandemic. Cities are colored as in panel a. Months are shown at the bottom of the figure, and each month is divided into 5-day intervals. New daily cases for the city of Khobar are shown on the Eastern Region plot. Major restrictions imposed by the Ministry of Health and by Royal decrees are indicated above plots. **c** Stacked bars showing the average numbers of new daily cases in sampling cities (Supplementary Note 1). **d** Estimate of effective reproduction number [Rt] over time in Saudi Arabia (top) and the estimate of effective population size [Ne], the relative population size required to produce the diversity seen in the sample (bottom). Central black lines show median estimates, and gray confidence areas denote the 95% credible intervals. The red horizontal red line represents an R of 1, the level required to sustain epidemic growth.

computationally intensive phylogeographic methods and additional population movement data.

**Origin of R203K/G204R SNPs.** A dated phylogeny of global samples showed that samples with the R203K/G204R SNPs are predominantly found in Nextstrain clades 20A, 20B, and 20C, and do not form a monophyletic group (Supplementary Fig. S4). Furthermore, a few samples are further found in the early appearing 19A and 19B clades. However, due to the limited number of mutations separating SARS-CoV-2 genomes constructing a reliable and robust phylogeny is problematic[19], and while different clades may be well supported, the exact relationship between clades is often less easily resolved. Although phylogenetic trees of SARS-CoV-2 genomes may appear to robustly reflect transmission events, collapsing branches with low support will typically result in extensive polytomies[20,21]. Additionally, the placement of individual virus genomes may be hampered by systematic errors, homoplasies, potential recombination, or co-infection of multiple virus strains[19,20,22–27]. It is therefore not clear if the phylogenetic distribution of samples with R203K/G204R SNPs reflects multiple independent origins of the SNPs, although it is evident that the R203K/G204R SNPs appeared early in the pandemic spread (Supplementary Fig. S4). Within our sampling window we observe an apparent transient increase in the frequency of R203K/G204R SNPs (Fig. 3a) in accordance with earlier observations[18,28]. In the global data, the peak in R203K/G204R frequency is slightly delayed compared to samples from Saudi Arabia. The initial global peak is observed in July 2020 followed by a decline until the fall of 2020, where the R203K/G204R SNPs once again increased along with the Spike protein Y501N mutation in the B1.1.17 lineage[17] (Fig. 3a).

**A mutant form of the nucleocapsid (N) protein associated with higher viral loads in COVID-19 patients in Saudi Arabia.** A genome-wide association study between SARS-CoV-2 SNPs and patient mortality identified the three consecutive SNPs (G28881A, G28882A, G28883C) underlying the R203K/G204R mutations (Fig. 3b, Supplementary Fig. S5). Of the 892 assembled genomes, 882 (98.9%) genomes either have the three reference alleles, GGG, or the three mutant alleles, AAC, at positions 28,881–28,883. This is similarly found in global samples deposited in GIASID in 2020, where 99.7% of samples with SNPs at positions 28,881-28,883 contain all three SNPs (Supplementary Fig. S6). In our samples, no other SNPs co-occur with the R203K/G204R SNPs (Supplementary Fig. S7). The frequency of the R203K/G204R SNPs is markedly higher in samples from Jeddah, where the observed frequency of 0.38 is more than 10-fold higher than the average of the other cities (Supplementary Table S1).

Using multivariable regression, we next evaluated the effect of the R203K/G204R SNPs on mortality, severity, and viral load in our COVID-19 patient samples for which a limited amount of clinical meta-datasets were available. Disease severity was defined as deceased patients and patients admitted to ICU.

For mortality and severity, we first fitted a logistic linear model using R203K/G204R SNPs as a covariate and adjusting by sex, age, comorbidities, hospital, and other SNPs. We included 12 additional SNPs (C241T, C1191T, C3037T, G10427A, C14408T, C15352T, C18877T, A23403G, G25563T, C26735T, T27484C, and C28139T) that co-occurred with the R203K/G204R SNPs in at least five samples in the model. The A23403G mutation results in the Spike protein D614G SNP that is associated with higher viral load[13]. Then, to adjust for confounding effects from time, we also fitted models that included time. In the models, age and time were included using smoothing splines to allow for potential non-linear relationships[29].

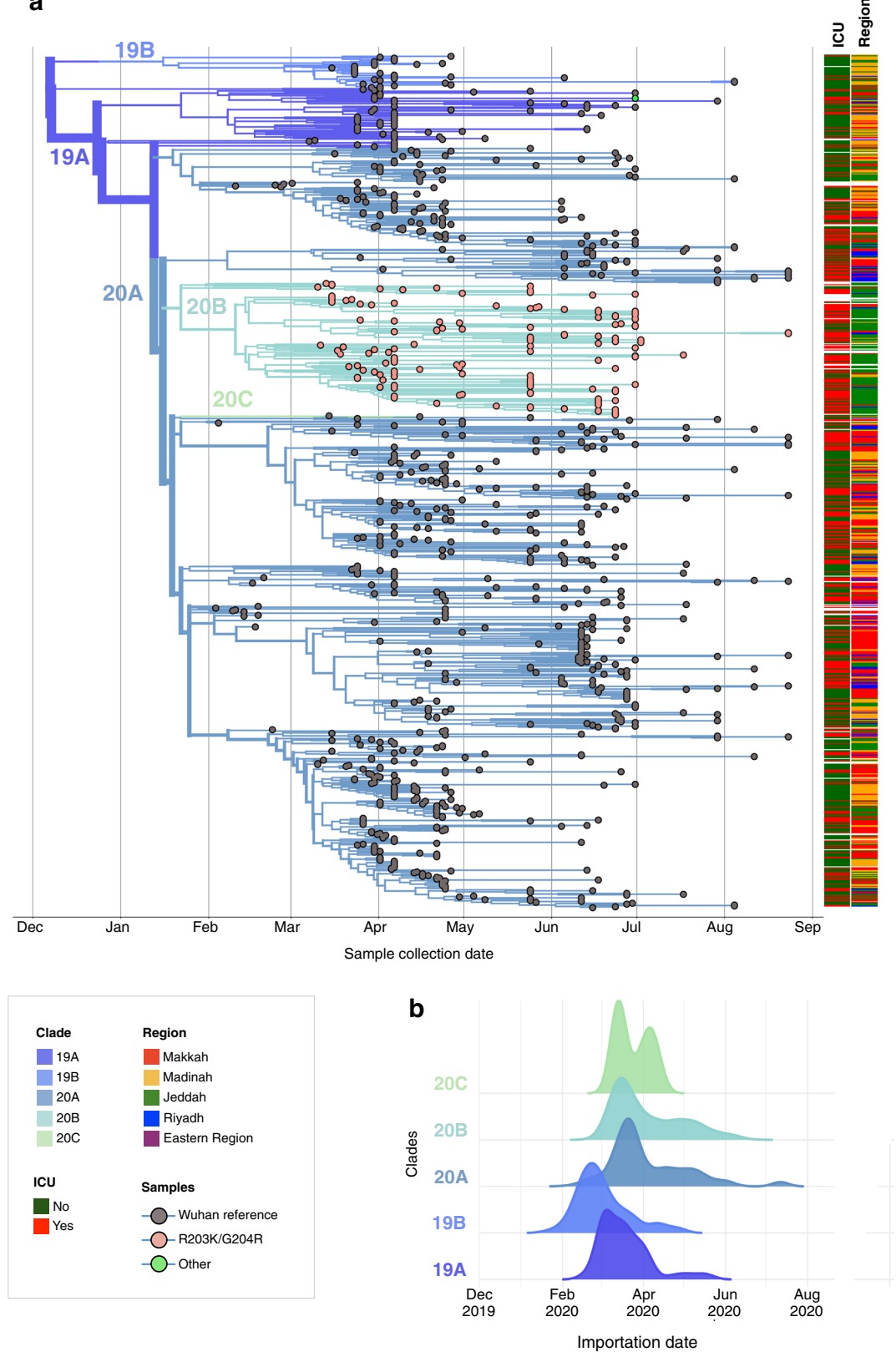

**Fig. 2 Phylodynamics of SARS-CoV-2 samples in Saudi Arabia. a** Global Time-scaled phylogeny of 952 Saudi samples colored by Nextstrain clades. Samples are shown as circles and colored according to their genotype at genome positions 28,881–28,883. Intensive Care Unit (ICU) status, patient outcome, and sampling region are indicated on the right of the tree. **b** Distributions of importation dates for the five Nextstrain (nextstrain.org) clades found in Saudi Arabia colored by clade.

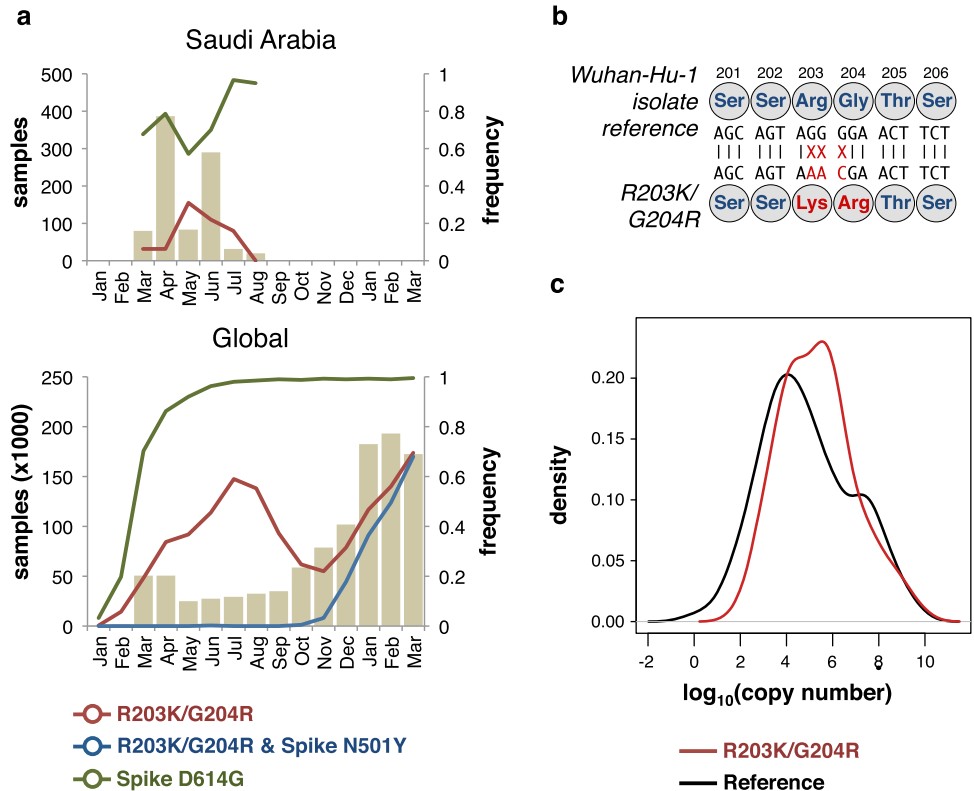

**Fig. 3 Higher viral loads in samples with R203K/G204R SNPs. a** Top: The numbers of samples from Saudi Arabia presented in this study are shown as bars by their sampling date (January 2020–March 2021). Bottom: Samples deposited in GISAID. On both plots, lines show the fraction of samples having the R203K/G204R SNPs (red line), having both the R203K/G204R SNPs and the Spike protein N501Y SNP (blue line), and having the Spike protein D614G SNP (green line). **b** Overview of the three SNPs underlying the N protein R203K/G204R changes. Amino acid numbers in the N protein are shown above. **c** Density distributions of virus copy numbers derived from Ct measurements. Ct values from the N1 primer pairs were normalized by RNase P primer pair values and converted to copy numbers from a standard curve. Only samples processed using the TaqPath™ kit (Thermofisher) were included (see Methods).

Using first a logistic regression model that did not include time, we observed a positive and statistically significant association between R203K/G204R SNPs and severity. Specifically, we found that the log-odds of severity increased by 1.18, 95% CI 0.22–2.13 (Supplementary Table S5). A positive significant association was also observed for the C14408T SNP, and a negative association for the C241T SNP (Supplementary Table S5).

In the time-adjusted model, the log-odds for the R203K/G204R SNPs increased to 1.38, 95% CI 0.28–2.48 (Supplementary Table S6). In this model, the C241T SNP again displayed a significant negative association, and a positive association was now observed for the C1887T SNP (Supplementary Table S6).

The relationship between mortality and R203K/G204R SNPs was positive and statistically significant in the model that did not include time with log-odds equal to 1.04, 95% 0.16–1.92. No significant association was observed for other SNPs (Supplementary Table S7). However, after adjusting for time as a variable, there was no longer any association between R203K/G204R SNPs and mortality (log-odds: 0.58, 95% CI −0.41–1.56) (Supplementary Table S8). The models thus suggest a temporal component in our observations, and it is important to note that the recorded mortalities from Jeddah are concentrated on just a few dates (Supplementary Fig. S8). One could speculate if temporal shifts in clinical settings in certain cases have impacted the balance between the severity of infection and mortality. Unfortunately, our dataset does not allow us to assess if the observed mortality rates are the result of shifts in treatment regimes or admission procedures during the sampling window.

We then tested if R203K/G204R SNPs were associated with higher viral copy numbers as indicated by the cycle threshold (Ct) values obtained through quantitative PCRs. As two different kits were used for the qPCR reactions (see Methods), we fitted adjusted models that besides sex, age, comorbidities, hospital, and time, included qPCR kits and the above-mentioned SNPs as covariates. From this adjusted regression we found a positive and statistically significant relationship between R203K/G204R SNPs and $\log_{10}$(viral copy number), with the mean of $\log_{10}$(viral copy number) values increasing by 1.33 units (95% CI 0.72–1.93) (Supplementary Table S9). Similarly, the model showed a positive significant association between the SNPs A23403G (Spike protein D614G) and C26735T SNPs and $\log_{10}$(viral copy number), the former being consistent with earlier reports[13,30]. A significant negative association was found for the C3037T C14408T, and G25563T SNPs (Supplementary Table S9). The positive and statistically significant association of R203K/G204R SNPs with higher viral load in critical COVID-19 patients (Fig. 3c) hence suggests their functional implications during viral infection.

**N mutant protein has high oligomerization potential and RNA-binding affinity.** The SARS-CoV-2 N protein binds the viral RNA genome and is central to viral replication[31]. Protein structure predictions have shown that the R203K/G204R mutations result in significant changes in protein structure[28], theoretically destabilizing the N structure[32], and potentially enhancing the protein's ability to bind RNA and alter its response to serine

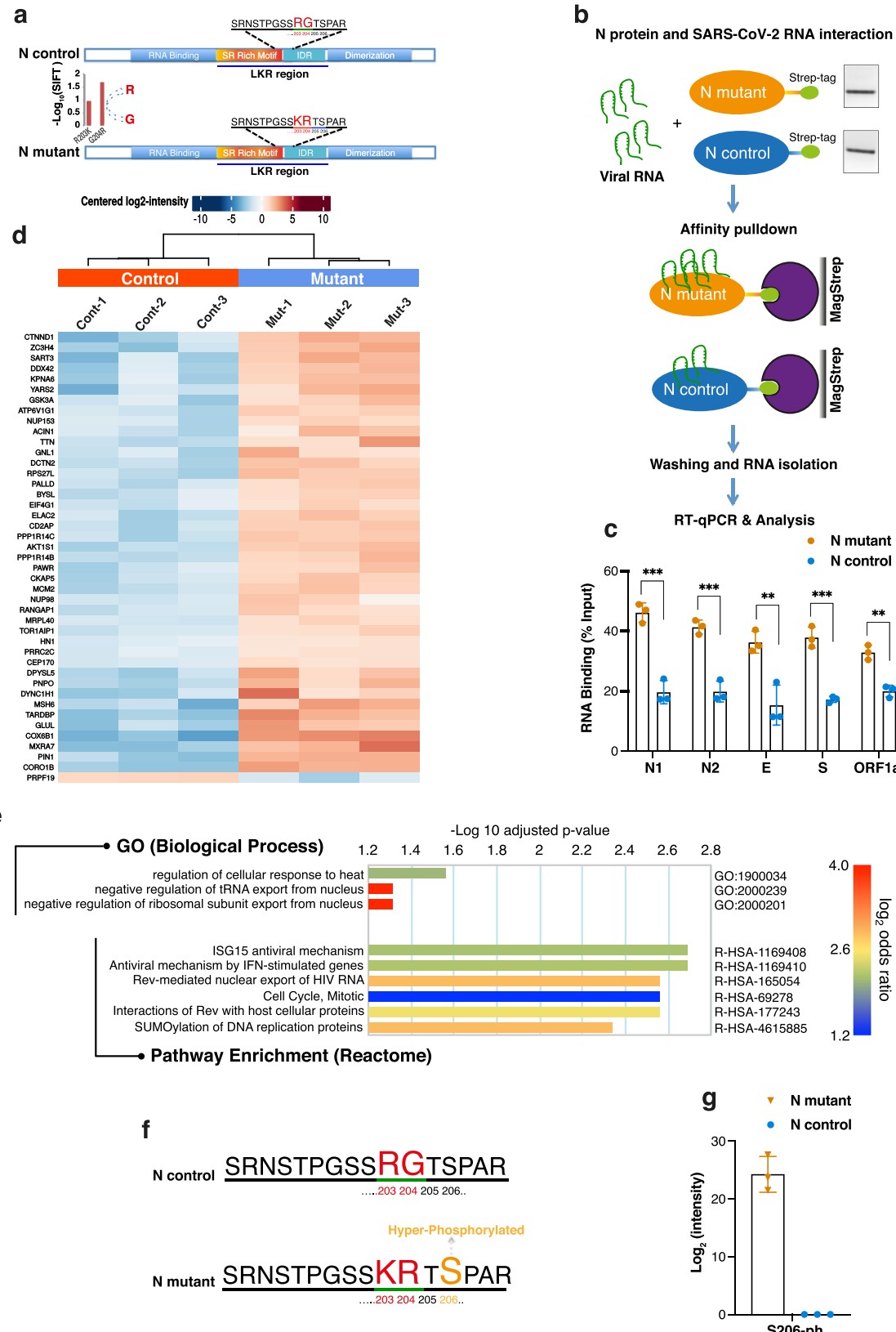

phosphorylation events[33]. The R203K/G204R mutations in the SARS-CoV-2 N protein are within the linkage region (LKR) containing the serine/arginine-rich motif (SR-rich motif) (Fig. 4a), known to be involved in the oligomerization of N proteins[34,35]. Protein cross-linking shows that N mutant protein (with the R203K/G204R mutations) has higher oligomerization

potential compared to the control N protein (without the changed amino acids) at low protein concentration (Supplementary Fig. S9).

Given that the oligomerization of N protein acts as a platform for viral RNA interactions[36], we sought to examine the binding affinity of mutant and control N protein with viral RNA isolated

**Fig. 4 RNA-binding and Affinity Purification Mass-Spectrometry (AP-MS) analysis of mutant and control SARS-CoV-2 N protein. a** A schematic diagram showing the SARS-CoV-2 N protein different domains (Upper: control, Lower: mutant) and highlighting the mutation site (R203K and G204R) and the linker region (LKR) containing a serine-arginine rich motif (SR-motif). The bar-plot (lower panel) indicates the SIFT[37] predicted deleteriousness score of substitution at position 203 and 204 from R to K and G to R respectively. **b** Sketch of In vitro RNA immunoprecipitation (RIP) procedure used for analysis of viral RNA interaction with mutant and control N protein (See methods for details). Isolated RNAs were analyzed by RT-qPCR using specific viral N gene (N1 and N2), E gene, S gene, and ORF1ab region. **c** Bar chart shows level of viral RNA retrieval (% input) with mutant and control N protein (± SD from $n = 3$ independent experiments, [two-sided t-test, p-values N1:0.00080 (***), N2:0.00088 (***), E:0.008 (**), S:0.00059 (***), and ORF1ab:0.002 (**)]). **d** Identification of host-interacting partners of mutant and control SARS-CoV-2 N protein by Affinity Mass-Spectrometry. Heatmap showing significantly differentially changed human proteins (3 replicates) interactome in mutant versus control N protein AP-MS analysis. **e** Gene Ontology (GO)-enrichment analysis of significantly changed terms between mutant and control proteins in terms of biological process and pathway enrichment. The scale shows p-value adjusted Log2 of odds ratio mutant versus control. **f** Profiling of phosphorylation status of mutant and control N protein by Mass-Spectrometry. Sketch showing part of SR-rich motif of SARS-CoV-2 N protein containing the KR mutation site (R203K and G204R) (Lower). The hyper-phosphorylated serine 206 (as shown in (**g**)) in the mutant N protein near the KR mutation site is indicated in orange color. **g** Phosphorylation status of mutant and control N protein was analyzed by mass spectrometry (±SD from $n = 3$ biologically independent experiments per affinity condition). Bar-plot shows the Log2 intensities of phosphorylated peptide (Serine 206) in control and mutant condition (see Supplementary Data 4).

from COVID-19 patient swabs. The RNA-binding activity of mutant and control N proteins was examined by pulled-down viral RNA through an in vitro RIP assay (Fig. 4b), and our data revealed that the mutant N protein enriched significantly higher level of viral RNA compared to control protein (Fig. 4c). This indicates a strong binding capability of mutant N proteins with viral RNA, which could potentially impact the essential roles of N protein at various stages of viral life cycle and its interaction with the host.

**The R203K/G204R mutations in the N protein affect its interaction with host proteins**. According to the SIFT tool[37], a substitution at position 204 from G to R in the N protein is predicted to affect functional properties (Fig. 4a). Therefore, we decided to investigate how the two amino acids substitution (R203K and G204R) in the N protein impact its functional interaction with the host that could modulate viral pathogenesis and rewiring of host cell pathways and processes. HEK-293T cells (three biological replicates) were used for affinity purification followed by mass spectrometry analysis (AP-MS) to identify host proteins associated with the control and mutant N protein (Supplementary Fig. S10). Protein identification was performed using MaxQuant software[38]. The compilation of the identified protein groups in mock and N protein (N control and mutant) AP-MS is presented in Supplementary Data 2 (Raw data are available via ProteomeXchange with identifier PXD027168). The majority (62%) of previously reported[39] N protein interacting partners overlapped with the identified unadjusted proteins list (Supplementary Fig. S10). We identified 43 human proteins that displayed significant (adjusted p-value ≤ 0.05, and Log2 fold-change ≥ 1) differential interactions with the mutant and control N protein (Fig. 4d, Supplementary Fig. S10, and see Supplementary Data 3 for both significant and non-significant differentially interacting protein list). Among these, 42 proteins showed increased interaction and one protein (PRPF19) showed decreased interaction with the N mutant (Fig. 4d and Supplementary Fig. S10). Among the group with increased interaction, we identified many proteins associated with TOR and other signaling pathways (such as AKT1S1 and PIN1), proteins associated with the viral process, viral transcription, and negative regulation of RNA nuclear export (NUP153 and NUP98), and proteins involved in apoptotic and cell death processes (PAWR and ACIN1) (Fig. 4d and Supplementary Fig. S10). We also identified proteins in the mutant condition that are linked with the immune processes and translation (Fig. 4d and Supplementary Fig. S10). Gene ontology analysis showed that the most enriched biological processes are associated with negative regulation of tRNA and ribosomal subunit export from the nucleus (Fig. 4e). This finding

suggests that the mutant virus may more efficiently inhibit and hijack the host translation to facilitate viral replication and pathogenesis. Further, many viruses can manipulate the host sumoylation process to enhance viral survival and pathogenesis[40]. By pathway enrichment analysis of differentially interacting proteins, we identified pathways associated with sumoylation and antiviral mechanisms (Fig. 4e).

**Serine 206 (S206) displays hyper-phosphorylation in the mutant N protein**. In SARS-CoV, it has been shown that phosphorylation of the N protein is more prevalent during viral transcription and replication[41] and inhibition of phosphorylation diminishes viral titer and cytopathogenic effects[42]. Recent elegant studies elaborated the role of N protein phosphorylation in modulating RNA binding and phase separation in SARS-CoV-2[36,43–45]. Thus, phosphorylation of N protein in the LKR region is critical for regulating both viral genome processing (transcription and replication) and nucleocapsid assembly[36,43]. To further understand the functional relevance of KR mutation in the N protein, we performed phosphoproteomic analysis in control and mutant conditions. We consistently found that the serine 206 (S206) site, which is next to the KR mutation site (Fig. 4f), is highly phosphorylated, specifically in the mutant N protein (Fig. 4g and Supplementary Data 4). Notably, the phosphorylation level at serine 2 (S2) and other serine sites (S79, S176, and S180) within the LKR region is detectable in both mutant and control conditions (Supplementary Data 4 shows only N protein phospho(STY) sites data. Complete raw data are available with the identifier PXD027168).

**The N mutant (R203K/G204R) induces overexpression of interferon-related genes in transfected host cells**. To understand whether the R203K/G204R mutations in the N gene affect host cell transcriptome, we transfected Calu-3 cells (4 biological replicates) with plasmids expressing the full-length N-control and N-mutant protein along with mock-transfection control. The transcriptome profile of N-mutant and N-control transfected cells displays a distinct pattern from the mock control (Supplementary Fig. S11). We identified 144 and 153 differentially expressed (DE) genes in the N-control and N-mutant transfected cells, respectively, with adjusted p-value < 0.05 and log2 fold-change ≥1 (Supplementary Fig. S11 and Supplementary Data 5). Among the DE genes, numerous interferon, cytokine, and immune-related genes are up-regulated, some of which are shown in Fig. 5 (for complete list see Supplementary Data 5). We found a robust overexpression of interferon-related genes in the N-mutant compared to N-control transfected cells (Fig. 5a–b) after

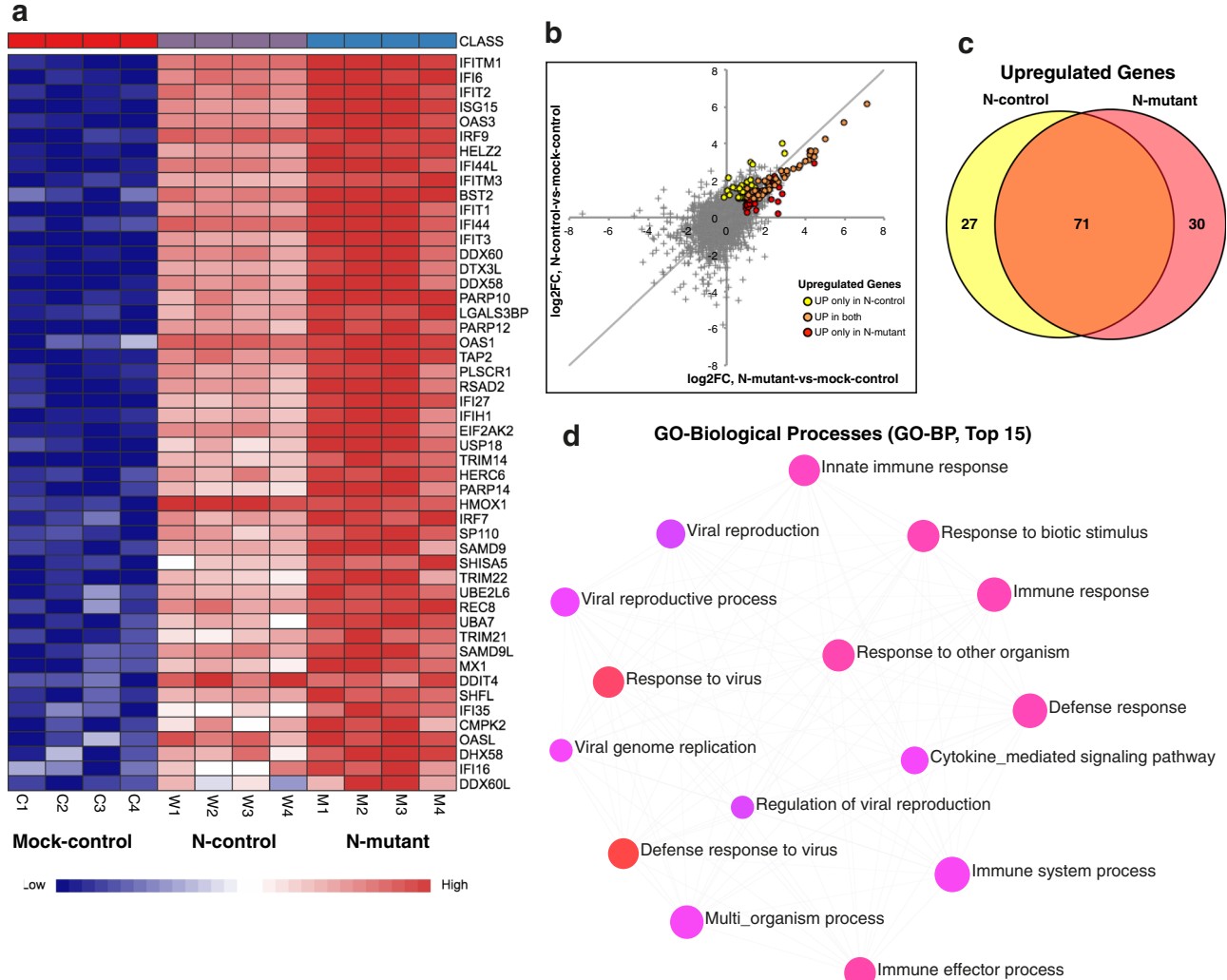

**Fig. 5 Transcriptional profiling of mutant and control N transfected cells.** Calu-3 cells were transfected with plasmids expressing the full-length N-control and N-mutant protein along with mock control (4 biological replicates per condition). 48-h post-transfection total RNA was isolated and subjected to RNA-sequencing using illumina NovaSeq 6000 platform. **a** Heatmap shows normalized expression of top significantly differentially expressed genes in N-mutant and N-control conditions (adj $p$-value <0.05 and log2 fold-change cutoff ≥1). Genes enriched in interferon and immune-related processes are overexpressed in the N-mutant transfected cells. The heatmap was generated by the visualization module in the NetworkAnalyst. **b** Plot showing comparison of fold-changes for all DE genes in N-mutant and N-control conditions. Differentially expressed genes display higher up-regulation in the N-mutant condition (as orange dots that represent common up-regulated genes are skewed towards the lower half of the diagonal). **c** Venn diagram shows the common and unique up-regulated genes in both conditions. **d** GO-enrichment analysis (top 15 pathways based on $p$-value and FDR are shown) of up-regulated genes. The enriched GO BP (Biological Processes) term is related to defense and interferon response. The enriched terms display an interconnected network with overlapping gene sets (from the list). Each node represents an enriched term and colored by its p-value from red to blue in ascending order (red shows the smallest $p$-value) as shown in Supplementary Data 6. The size of each node corresponds to number of linked genes from the list.

adjusting for fold-change (Supplementary Fig. S11). Indeed, strong overexpression of interferon and chemokine-related genes (Supplementary Data 5) were reported in critical COVID-19 patients[46,47]. Recent reports further indicate a link between increased expression of interferon-related genes and higher viral load in severe COVID-19 patients[48–50]. Also, we found over-expression of other genes such as ACE2, STAT1[47], and TMPRSS13[51] (Fig. 5a and Supplementary Data 5) that are elevated in critical COVID-19 disease.

Pathway enrichment analysis (top 15 pathways based on p-value and FDR) of the up-regulated genes (Fig. 5c) shows an over-representation of biological processes associated with response to the virus (Fig. 5d and Supplementary Data 6). Similarly, all DE genes were related to substantially enriched pathways, such as interferon-related response, cytokine production, and viral reproductive

processes (Supplementary Fig. S11). The enriched GO terms display an interconnected network highlighting the relationships between up-regulated overlapping genes sets in these pathways (Fig. 5d and Supplementary Fig. S11). Taken together, these results suggest that the R203K/G204R mutations in the N protein may enhance its function in provoking a hyper-expression of interferon-related genes that contribute to the cytokine storm in exacerbating COVID-19 pathogenesis.

## Discussion
From 892 samples collected across the country over the course of approximately 6 months we have analyzed the dynamics of transmission and diversity of SARS-CoV-2 in Saudi Arabia. The lineage analysis of assembled genomes highlights the repeated

influx of SARS-CoV-2 lineages into the Kingdom. The earliest estimated importation dates point to an entry during the early stages of the pandemic (Fig. 2b). From estimates of viral genetic diversity and reproduction rate, we find that decreased diversity and reproduction rate coincides with imposed national curfews and is followed by an observed drop in reported COVID-19 cases (Fig. 1c, d).

Our COVID-19 patient data allowed to us detect three SNPs— underlying the N protein R203K and G204R mutations—significantly associated with higher viral load. It is worth noting that two studies have found higher viral load has in infected patients to be associated with severity and mortality[52,53]. Among our samples we initially observed an apparent association between the R203K/G204R mutations and mortality, however, the association was no longer statistically significant when correcting for sampling time. The association of N protein R203K and G204R mutations with higher viral load persists after adjusting for time but not with mortality, suggesting that the mortality rate of severe infections may be influenced by other factors such as changes in treatment regimes as well as the complexities of host response to SARS-CoV-2 infection. Unfortunately, the available data do not allow a further assessment of this.

The N protein of SARS-CoV-2, an abundant viral protein within infected cells, serves multiple functions during viral infection, which besides RNA binding, oligomerization, and genome packaging, playing essential roles in viral transcription, replication, and translation[31,54]. Also, the N protein can evade immune response and perturbs other host cellular processes such as translation, cell cycle, TGFβ signaling, and induction of apoptosis[55] to enhance virus survival. The critical functional regulatory hub within the N protein is a conserved serine-arginine (SR) rich-linker region (LKR), which is involved in RNA and protein binding[56], oligomerization[34,35], and phospho-regulation[36,43].

We show that the mutant N protein-containing R203K and G204R changes has higher oligomerization and stronger viral RNA-binding ability, suggesting a potential link of these mutations with efficient viral genome packaging. The R203K and G204R mutations are in close proximity to the recently reported RNA-mediated phase-separation domain (aa 210–246)[45] that is involved in viral RNA packaging through phase separation. This domain was thought to enhance phase separation also through protein–protein interactions[45]. Further studies are needed to examine any definite link between KR mutation and phase separation; however, the differential interaction of host proteins, as shown in our study could affect this process.

Moreover, the functional activities of the N protein at different stages of the viral life cycle are regulated by phosphorylation-dependent physiochemical changes in the LKR region[43]. Although all individual phosphorylation sites may not be functionally important[33,57], the specific enhancement of phosphorylation at serine 206 in the mutant N protein shown in this study hints at its functional significance. The serine 206 can form a phosphorylation-dependent binding site for protein 14-3-3, involved in cell cycle regulatory pathways regulating human and virus protein expression[58]. Multiple lines of evidence show that N protein phosphorylation is critical for its dynamic localization and function at replication-transcription complexes (RTC), where it promotes viral RNA transcription and translation by recruiting cellular factors[41–43,59–62]. The enrichment of glycogen synthase kinase 3 A (GSK3A) with the mutant N protein, could specifically phosphorylate serine 206 in the R203K/G204R mutation background. GSK3 was shown to be a key regulator of SARS-CoV replication due to its ability to phosphorylate N protein[42]. Phosphorylation of serine 206 acts as priming site for initiating a cascade of GSK-3 phosphorylation events[42,43]. Also, GSK3 inhibition dramatically reduces the production of viral particles and the cytopathic effect in SARS-CoV-infected cells[42]. The detection of hyperphosphorylation at serine 206 only in the N-mutant could also point toward the phosphorylation timing difference or more phosphorylation stability at this site in the mutant background. Finally, our analysis of the transcriptome in transfected cells suggests that the mutant N protein induce overexpression of interferon-related genes that can aggravate the viral infection by inducing cytokine storm.

As the COVID-19 pandemic is still ongoing, there is a need for novel therapeutic strategies to treat severe infections in patients. Our identified interaction pathways and inhibition of serine 206 phosphorylation could be used as potential targets for therapies.

In conclusion, our results suggest a major influence of the R203K/G204R mutations on the essential properties and phosphorylation status of SARS-CoV-2 N protein that may lead to increased host response and heightened efficacy of viral infection.

## Methods

**Sample collection**. As part of the study, nasopharyngeal swab samples were collected in 1 ml of TRIzol (Ambion, USA) from 892 COVID-19 patients with various grades of clinical disease manifestations—consisting of severe, mild and asymptomatic symptoms. The anonymized samples were amassed from 8 hospitals and one quarantine hotel located in Madinah, Makkah, Jeddah, and Riyadh (Supplementary Table S1). Patient metadata in the form of age, sex, comorbidities, ICU submission, and mortality were provided by the hospitals (Supplementary Table S2) and used for statistical analysis. Ethical approvals were obtained from the Institutional review board of the Ministry of Health in the Makkah region with the numbers H-02-K-076-0420-285 and H-02-K-076-0320-279, as well as the Institutional review board of Dr. Sulaiman Al Habib Hospital number RC20.06.88 for samples from Riyadh and the Eastern regions respectively.

**RNA Isolation**. RNA was extracted using the Direct-Zol RNA Miniprep kit (Zymo Research, USA) following the manufacturer's instructions, along with several optimization steps to improve the quality and quantity of RNA from clinical samples. The optimization included extending the TRIzol incubation period, and the addition of chloroform during initial lysis step to obtain the aqueous RNA layer. The quality control of purified RNA was performed using Broad Range Qubit kit (Thermo Fisher, USA) and RNA 6000 Nano LabChip kit (Agilent, USA) respectively. RT-PCR was conducted using the one-step Super Script III with Platinum Taq DNA Polymerase (Thermo Fisher, USA) and TaqPath COVID-19 kit (Applied Biosystems, USA) on the QuantStudio 3 Real-Time PCR instrument (Applied Biosystems, USA) and 7900 HT ABI machine. The primers and probes used were targeting two regions in the nucleocapsid gene (N1 and N2) in the viral genome following the Centre for Disease Control and prevention diagnostic panel, along with primers and probe for human RNase P gene (CDC; fda.gov/media/134922/download) (Supplementary Table S10). Samples were considered COVID positive once the cycle threshold (Ct) values for both N1 and N2 regions were less than 40. For amplicon seq purposes, the samples chosen were of Ct less than 35 to ensure successful genome assembly in order to upload on GISAID.

**Sequencing and data analysis**. cDNA and amplicon libraries were prepared using the COVID-19 ARTIC-V3 protocol, producing ~ 400 bp amplicons tiling the viral genome using V3 nCoV-2019 primers (Wellcome Sanger Institute, UK; dx.doi.org/10.17504/protocols.io.beuzjex6). Amplicons were then processed for deep, paired-end sequencing with the Novaseq 6000 platform on the SP 2 ×250 bp flow cell type (Illumina, USA).

**Genome assembly, SNP and indel calling**. Illumina adapters and low-quality sequences were trimmed using Trimmomatic (v0.38)[63]. Reads were mapped to SARS-CoV-2 Wuhan-Hu-1 NCBI reference sequence NC_045512.2 using BWA (v0.7.17)[64]. Mapped reads were processed using GATK (v4.1.7) pipeline commands MarkDuplicatesSpark, HaplotypeCaller, VariantFiltration, SelectVariants, BaseRecalibrator, ApplyBQSR, and HaplotypeCaller to identify variants[65]. High quality SNPs were filtered using the filter expression:
"QD < 2.0 || FS > 60.0 || SOR > 3.0 || MQRankSum < −12.5 || ReadPosRankSum < −8.0"
High quality Indels were filtered using the filter expression:
"QD < 2.0 || FS > 200.0 || SOR > 10.0 || ReadPosRankSum < -20.0"
Consensus sequences were generated by applying the good quality variants from GATK on the reference sequence using bcftools (v1.9) consensus command[66]. Regions which are covered by less than 30 reads are masked in the final assembly with 'N's.

Consensus assembly sequences were deposited to GISAID (Supplementary Data 1)[11]. To retrieve high-confidence SNPs assembled sequences were re-aligned

against the Wuhan-Hu-1 reference sequence (NC_045512.2), and only positions in the sample sequences with unambiguous bases in a 7-nucleotide window centered around the SNP position were kept for further analysis.

**Phylogenetic analysis.** To generate the phylogeny of Saudi samples with a global context, a total of 308,012 global sequences were downloaded from GISAID on 31 December 2020, filtered and processed using Nextstrain pipeline[12]. Global sequences were grouped by country and sample collection month and 20 sequences per group were randomly sampled which resulted in 10,873 global representative sequences and 952 Saudi sequences. The phylogeny was constructed using IQ-TREE (v2.0.5)[67], clades were assigned using Nextclade and internal node dates were inferred and sequences pruned using TreeTime (v0.7.5)[68]. Nextstrain protocol was followed for the above-mentioned steps. The resulting global phylogenetic tree was reduced to retain the branches that lead to Saudi leaf nodes and visualized using baltic library (https://github.com/evogytis/baltic).

**Phylodynamic analysis.** Phylodynamic analyses use the same sequence subset used in the full phylogenetic analysis, extracted from the GISAID SARSCoV-2 database[11]. Wrapper functions for the importation date estimates and skygrowth model are provided in the sarscov2 R package as 'compute_timports' and 'skygrowth1' respectively (https://github.com/JorgensenD/sarscov2Rutils)[69].

**Importation date estimates for Nextstrain clades.** Importation rate estimates were carried out using all available sequence data for Saudi Arabia deposited on GISAID[11] up to 31 December 2020, including the sequences described in this paper. Sequences were grouped by Nextstrain clade for analysis using the Nextstrain_clade parameter[12] in the GISAID metadata table. Additional international sequences were selected for each of the included Nextstrain clade based on Tamura Nei 93 distance with the C program tn93 (v1.0.6) (github.com/veg/tn93)[70]. Five hundred sequences were selected from available closely related sequences in a time stratified manner, taking every N/500th sequence from the set of N sequences arranged by date, rounded to the nearest integer.

For each Nextstrain clade a maximum-likelihood phylogeny was produced with IQTree (v1.6.12) with an HKY substitution model[67,71]. These trees were dated using the R package treedater (v0.5.0) after collapsing short branch lengths and resolving polytomies randomly fifteen times for each clade with the functions di2multi and multi2di from the *ape* R package (v5.5)[72,73]. A strict molecular clock was used when estimating dated phylogenies, constrained between 0.0009 and 0.0015 substitutions per site per year[74]. The state of each internal node in the phylogeny was reconstructed by maximum parsimony with the R package phangorn (v2.7.0)[75]. As this method cannot directly estimate the timing of an importation event, importations were estimated to occur at the midpoint of a branch along which a state change occurs between the internal and external samples.

The probability density of importation events over time into Saudi Arabia by cluster is presented in Fig. 2b where a gaussian kernel density estimator is used with a bandwidth of 5.51 chosen using the Silverman rule of thumb, implemented in the geom_density function of the ggplot2 R package (v3.3.5)[76,77].

**Skygrowth model.** The assembled sequences, together with all other reported sequences for Saudi Arabia available on GISAID on 31 December 2020, were used to construct estimates of the effective population size and growth rate of SARS-CoV2 in Saudi Arabia over the course of the first wave of the epidemic (March to September 2020)[11]. The collected sequences were used to produce a maximum likelihood phylogeny with an HKY substitution model in IQTree (v1.6.12)[67,71]. A set of 1000 bootstrap pseudo-replicate trees were produced with the ultrafast bootstrap approximation[78]. For each bootstrap phylogeny, branches with length less than $10^{-5}$ were collapsed and polytomies were resolved randomly using the di2multi and multi2di functions in the ape R package (v5.5)[72]. These dichotomous trees were produced for subsequent molecular clock analysis and coalescent analysis. The set of 1000 initial bootstrap trees and 1000 additional phylogenies with randomly resolved short branches were dated using the reported collection dates for the sequenced samples as the date of the corresponding tip in each phylogeny. A strict molecular clock was used when estimating dated phylogenies, constrained between 0.0009 and 0.0015 substitutions per site per year with the R package treedater (v0.5.0)[73,74].

The R package Skygrowth (v0.3.1) was used with these phylogenies to estimate the growth rate and effective population size of SARS-CoV-2 in Saudi Arabia over time[16]. Skygrowth is a Bayesian non-parametric model of effective population size, with the primary difference to other skyline smoothing methods being that the first-order stochastic process is defined in terms of the growth rate of the effective population size ($Ne$) rather than $Ne$ itself. The growth rate of $Ne$ often has a simple relationship with the growth rate of the epidemic from which samples are collected. Here, under the assumption of a simple SIR transmission process, we return the effective reproduction number as in Eq. (1) where $\dot{Ne}_{(t)}$ is the derivative of $Ne_{(t)}$,

$(\dot{Ne}_{(t)}/Ne_{(t)})$ is the growth rate of $Ne$, and $\psi$ is the mean generation time.

$$R(t) \approx 1 + \psi\left(\dot{Ne}_{(t)}/Ne_{(t)}\right) \qquad (1)$$

The model included 35 timesteps and an exponential prior on the smoothing parameter tau (precision) corresponding to a 1% change in growth per week. The growth rate output was converted to an estimate of R over time using an assumed generation time $\psi$ of 9.5 days[79]. The model is fitted from March 5th to ensure exponential growth is established in the population prior to model fitting as we cannot estimate the growth rate or effective reproduction number when virus circulation is driven purely by importation events.

Although phylogenetic methods are sensitive to sampling rate changes over time, the coalescent method used in Skygrowth is relatively robust to heterogenous sampling through time, but can still be biased by unequal sampling in space or risk groups. Computation of reproduction numbers is premised on equivalence of the growth rate of the epidemic and the growth rate of the effective population size, which does not hold when the transmission rate is highly variable.

**Origin of R203K/G204R SNPs.** A total of 590 K samples submitted to GISAID until February 24 were downloaded and SNPs indentifed by mapping against the Wuhan-Hu-1 reference sequence (NC_045512.2) using minimap2 (v2.17)[80]. The variants were queried to count the distribution of triplets among various Nextstrain clades (Supplementary Fig. S6). To identify if there are lineages of triplet SNPs in clades other than 20B, a phylogenetic tree was constructed by including all R203K/G204R samples found in other clades outside 20B and its subclades (Supplementary Fig. S4). As it was already evident that 20B and its subclades contains lineages of R203K/G204R samples, subsamples from 20B and its subclades were sufficient to obtain a total of 16,386 samples.

**Genome-wide associations.** Tests were carried out as a conventional case-control setup, and for each SNP a contingency table was constructed as (deceased patients with SNP/ deceased patients without SNP)/(non-deceased patients with SNP/ non-deceased patients without SNP). P-values were calculated using Fisher's exact two-tailed test.

**Allele frequency estimate.** Duplicate reads were removed from the mapped short Illumina sequence reads using picard tools' MarkDuplicates function (v2.20.4, Broad Institute, GitHub repository. http://broadinstitute.github.io/picard/). Frequencies were collected directly from the output of samtools (v1.8) mpileup[66]. Distal read positions were excluded and only read mapping and base calling qualities of at least 30 were considered.

**Multivariable regression analysis.** Statistical analyses were performed with the statistical software R version 4.0.3[81] and the R package mgcv (v1.8.33). Models for mortality and severity used 892 observations (all data).

**Plasmid and cloning.** The pLVX-EF1alpha-SARS-CoV-2-N-2xStrep-IRES-Puro was a gift from Nevan Krogan (Addgene plasmid # 141391; RRID:Addgene_141391)[39]. The three consecutive SNPs (G28881A, G28882A, G28883C), corresponding to N protein mutation sites R203K and G204R, were introduced by megaprime PCR mutagenesis using the primers listed in Supplementary Table S10.

**Cell culture and transfection.** HEK293T (ATCC; CRL-3216) cells were grown in Dulbecco's modified Eagle's medium (DMEM) (4.5 g/l d-glucose and Glutamax, 1 mM sodium pyruvate) (GIBCO) and 10% fetal bovine serum (FBS; GIBCO) with penicillin–streptomycin supplement, according to standard protocols (culture condition 37 °C and 5% $CO_2$). Calu-3 (ATCC HTB-55) cells were grown in DMEM (1.0 g/l d-glucose, 2mM L-glutamine, 1 mM sodium pyruvate) with addition of 1% non-essential amino acids, 1500 mg/L sodium bicarbonate, and 10% fetal bovine serum (FBS; GIBCO). Transfection of ten million cells per 15-cm dish with 2XStrep-tagged N plasmid (20ug/transfection) was performed using lipofectamine-2000 according to standard protocol. For Calu-3 cells, double transfection was conducted; the first transfection was done on the day of cell splitting and the second after 24 h of culture.

**Affinity purification and on-bead digestion.** Cell lysis and affinity purification with MagStrep beads (IBA Lifesciences) were manually performed according to the published protocol[39] with minor modifications. Briefly, after transfection (48 h) cells were collected with 10 mM EDTA in 1xPBS and washed twice with cold PBS (samples used for phosphorylation analysis were collected and washed in the presence of phosphatase inhibitor cocktails). The cell pellets were stored at −80 °C. Cells were lysed in lysis buffer (50 mM Tris-HCl, pH 7.4, 150 mM NaCl, 1 mM EDTA, 0.5% NP40, supplemented with protease and phosphatase inhibitor cocktails) for 30 min while rotating at 4 °C and then centrifuge at high speed to collect the supernatant. The cell lysate was incubated with prewashed MagStrep beads (30 μl per reaction) for 3 h at 4 °C. The beads were then washed four times with wash buffer (50 mM Tris-HCl, pH 7.4, 150 mM NaCl, 1 mM EDTA, 0.05% NP40,

supplemented with protease and phosphatase inhibitor cocktails) and then proceed with on-bead digestion. The on-bead digestion was carried out as described before[39]. Briefly, after the final wash the beads were washed once with exchange buffer (50 mM Tris-HCl PH 7.5, 100 mM NaCl). Bead-bound proteins were alkylated with 3 mM iodoacetamide in the dark for 45 min and then quenched with 3 mM DTT for 15 min. Digestion was performed using 1 μl trypsin (1 μg/μl, Promega) overnight with shaking (1000 rpm) and the peptides were then purified. For affinity confirmation, bound proteins were eluted using buffer BXT (IBA Lifesciences) and after running on SDS-PAGE were subjected to silver staining and western-blot using anti-strep-II antibody (ab76949) (dilution used 1:1000). To purify clean 2xStrep-tagged N protein (mutant and control), we applied stringent washing and double elution strategy.

**MS analysis using Orbitrap Fusion Lumos**. The MS analysis was performed as described previously[82,83] with slight modifications. Briefly, approximately 0.5 μg of peptide mixture in 0.1% formic acid (FA) was injected into a nanotrap (PepMap 100, C18, 75 μm × 20 mm, 3 μm particle size) and desalted for 5 min with 0.1% FA in water at a flow rate of 5 μl/min. They were then eluted and analyzed using an Orbitrap Fusion mass spectrometer (MS) (Lumos, Thermo Fisher Scientific) coupled with an UltiMate™ 3000 UHPLC (Thermo Scientific). The peptides were separated by an EasySpray C18 column (50 cm × 75 μm ID, PepMap C18, 2 μm particles, 100 Å pore size, Thermo Scientific) with a 75-min gradient at constant 300 nL/min, at 40 °C. The electrospray potential was set at 1.9 kV, and the ion transfer tube temperature was set at 270 °C. A full MS scan with a mass range of 350–1500 $m/z$ was acquired in the Orbitrap at a resolution of 60,000 (at 400 $m/z$) using the profile mode, a maximum ion accumulation time of 50 ms, and a target value of 2e5. The most intense ions that were above a 2e4 threshold and carried multiple positive charges (2–6) were selected for fragmentation (MS/MS) via higher energy collision dissociation (HCD) with normalized collision energy at 30% within the 2 s cycling time. The dynamic exclusion was 30 s. The MS2 was acquired with data type as centroid at a resolution of 30,000. Protein identification analysis from the raw mass spectrometry data was performed using the Maxquant software (v1.5.3.30)[38] as described[82].

For phosphorylated peptides, we used Maxquant label-free quantification (LFQ)[38]. The analysis and quantification of phosphorylated peptides were performed according to published protocol[84]. The sample file naming format and the datasets associated with the MaxQuant analysis are represented in Supplementary Table S11.

**Analysis of differential interaction**. First, the identified protein group data were corrected for non-specific background binding by removing all proteins detected in mock control (cells transfected with the plasmid vector without N gene) affinity mass spectrometry (Supplementary Data 7). The normalized LFQ data were processed for statistical analysis on the LFQ-Analyst a web-based tool[85] to performed pair-wise comparison between mutant and control N protein AP-MS data. The significant differentially interacting proteins between mutant and control conditions were identified. The threshold cutoff of adjusted p-value < = 0.05, and Log fold-change > = 1 were used. Among the replicates, outliers were removed based on correlation and PCA analysis. The GO-enrichment analysis was performed on the LFQ-Analyst[85].

**BS³ cross-linking**. Bis(sulfosuccinimidyl) suberate (BS³, Thermo Scientific Pierce) was used for cross-linking of control and mutant N protein to analyse the oligomerization properties. The experiment was performed as reported previously[86]. Briefly, 2xStrep-tagged N protein (mutant and control) was purified as mentioned above. The purified N protein (mutant and control) were cross-linked using 2 mM Bis(sulfosuccinimidyl) suberate (BS³) for 30 min at room temperature. The control and cross-linked forms of N proteins (mutant and control) were separated on SDS-PAGE, subjected to silver staining and densitometry analysis of bands. GraphPad Prism (v9.1.1) was used for analysis and graph generation.

**In vitro protein–RNA interaction (RIP) assay**. The in vitro interaction assay was performed using purified 2xStrep-tagged N protein (mutant and control) and total isolated RNA from patient swabs as mentioned above in the RNA isolation section. The 2xStrep-tagged (in bead-bound condition) N protein (mutant and control) was incubated with total RNA in reaction buffer (50 mM Tris HCl pH7.5, 150 mM KCl, 0.1 mM EDTA, 1 mM DTT, 5% Glycerol, 0.02% NP40, 1 mM PMSF, 40 U RNase OUT™). After shaking incubation for 45 min, N proteins (mutant and control) were pulldown using MagStrep beads on a magnetic separator with four washes (using the same reaction buffer). After the final wash, the RNA was isolated by adding Trizol using the Zymo direct-zol method. The isolated RNAs were analyzed by RT-qPCR using specific viral N gene (N1 and N2), E gene, S gene, and ORF1ab region primers (Supplementary Table S10). GraphPad Prism (v9.1.1) was used for analysis and graph generation.

**RNA-sequencing and differential gene expression analysis**. Calu-3 cells were transfected with plasmids expressing the full-length N-control and N-mutant protein along with mock control. After 48-h cells were harvested in Trizol and total RNA was isolated using Zymo-RNA Direct-Zol kit (Zymo, USA) according to the

manufacture's instruction. The concentration of RNA was measured by Qubit (Invitrogen), and RNA integrity was determined by Bioanalyzer 2100 system (Agilent Technologies, CA, USA). The RNA was then subjected to library preparation using Ribozero-plus kit (Illumina). The libraries were sequenced on NovaSeq 6000 platform (Illumina, USA) with 150 bp paired-end reads.

The raw reads from Calu-3 RNA-sequencing were processed and trimmed using trimmomatic[63] and mapped to annotated ENSEMBL transcripts from the human genome (hg19)[87,88] using kallisto (v0.43.1)[89]. Differential expression analysis was performed after normalization using EdgeR integrated in the NetworkAnalyst[90]. GO biological process and pathway enrichment analyses on differentially expressed genes were performed using NetworkAnalyst[90].

**Reporting summary**. Further information on research design is available in the Nature Research Reporting Summary linked to this article.

## Data availability

Supplementary Information is available for this paper. Assembled virus genomes are available at GISAID (Supplementary Data 1). Raw viral reads, assembled virus genomes, and reads from RNA-Seq analysis of transfected Calu-3 cells have been uploaded to European Nucleotide Archive under the Study accession number PRJEB45515. The affinity-purification mass spectrometry (AP-MS) proteomics data have been deposited to the ProteomeXchange Consortium via the PRIDE partner repository[91] with the dataset identifier PXD027168. Source data are provided with this paper.

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

## Acknowledgements

The authors are sincerely grateful to all hospital members for providing samples and collating metadata in such an unprecedented pandemic, along with the MOH and ethical committee, which rendered it permissible. We thank the KAUST Rapid Research Response Team (R3T) under the Vice President Research (VPR) office in KAUST for generous financial support. We also thank Erik Talley from KAUST Health Safety and Environment (HSE) and Hani Bukhari from KAUST Security for providing timely logistical support for samples transport during COVID-19 Curfew restrictions in the Kingdom. We extend our thanks and appreciation to GDRS director, PH. Athari Alotaibi (General Director for Research and Studies, MOH) for her vigorous facilitation of the research project, and Mohammad Fawzi (General Directorate of Health Affairs) for his help with the metadata collection. We also deeply thank Dr. Wael Hamzah Motair, Dr. Nader Hamzah Motair, Dr. Hatim Khogeer and the General Directorate of Health Affairs of Makkah Region (GDHAMR), MOH for all their help and assistance to our study. We thank Sebastian Gornik (University of Heidelberg) for critical comments on the preprint version of this manuscript. We gratefully acknowledge all of the authors from the originating laboratories responsible for obtaining the specimens and the submitting laboratories where genetic sequence data were generated and publicly shared via the GISAID Initiative, on which was partially used for additional support for some of the conclusions drawn in this study. This work was supported by the following grants: KAUST Rapid Research Response Team (R3T) by Vice President Research (VPR) office in KAUST. KAUST faculty baseline fund (BAS/1/1020-01-01) to AP. KACST Grants, Proposal number: 5-20-01-002-0008. MOH COVID-19 project grants number 341. MOH COVID-19 project grants number 754. The deputyship for Research and Innovation, Ministry of Education in Saudi Arabia, project number (436) to AMH. This project was conducted under the IRB approvals of the MOH (H-02-K-076-0420-285), KAUST (20IBEC14), and at the Dr. Suliman Al-Habib Medical group (HAP-01-R-082) in KSA.

## Author contributions

A.P. conceived the study and directed the work and acquired funding from KAUST and supplemental funding from King Abdulaziz City for Science and Technology (KACST). A.P., T.M., M.S., S.H. and S.M. designed the research. IRB and ethical approvals from MOH were acquired by A.K., A.H., N.A., A.M. and S.H. to cover the collection from several cities in the Kingdom. S.H. and A.K. acquired funding from the Saudi Ministry of Health (MOH) numbers 754 and 341, utilized in the study. S.H. organized and directed sample collection and metadata collections with aid from F.A., A.S., A.O., S.S., J.T., A.A., N.K., K.K., K.A. and A.D.; S.M. directed the wet lab work involving sample reception, metadata record-keeping, RNA extraction, quality control, and library preparation, with aid from A.S., F.B., R.S., M.S., A.O., L.E., O.D., S.H. and R.N.; Illumina sequencing runs were performed by S.M., L.E., S.P. and I.R.; Genome assemblies and submission to GISAID was done by R.N.; Phylogenetic and lineage analysis was done by R.N., Q.G., D.J. and E.V.; In-depth SNP data analysis was performed by T.M.; Statistical analysis done by P.E.M. and E.V.; Functional validation of this link was established by M.S.; Wet lab experiments and data analysis including affinity mass spectrometry, RNA interaction, host-cell transcriptome analysis was performed by M.S.; T.M. wrote the initial draft of the manuscript with input from M.S., S.M., S.H., R.N. and Q.G., followed by edits from A.P.; The final version was produced by T.M., M.S. and A.P. after input from all co-authors. F.A., R.N., D.J., A.S., A.S., F.B., Q.G., R.S. co-second authors contributed equally to this work. A.O., L.E., O.D., R.N. co-third authors contributed equally to this work. S.P., H.Z., I.R. co-fourth authors contributed equally to this work.

## Competing interests

The authors declare no competing interests.

## Additional information

[1]King Abdullah University of Science and Technology (KAUST), Pathogen Genomics Laboratory, Biological and Environmental Science and Engineering (BESE), Thuwal-Jeddah 23955-6900, Saudi Arabia. [2]Infectious Disease Research Department, King Abdullah International Medical Research Centre, Ministry of National Guard Health Affairs, Jeddah, Saudi Arabia. [3]King Saud bin Abdulaziz University for Health Sciences, Ministry of National Guard Health Affairs, Jeddah, Saudi Arabia. [4]Infectious Diseases Department, King Fahad Hospital, Madinah, MOH, Saudi Arabia. [5]Infectious Diseases Department, King Abdullah Medical Complex, Jeddah, MOH, Saudi Arabia. [6]School of Public Health, Faculty of Medicine, Imperial College, Norfolk Place, St Mary's Campus, London, United Kingdom. [7]Dr. Suliman Al-Habib Medical Group, Riyadh, Saudi Arabia. [8]Department of Neuroscience, King Faisal Specialist Hospital and Research Center, Riyadh, Saudi Arabia. [9]Department of Preventive Medicine, Ministry of National Guard - Health Affairs, Riyadh, Saudi Arabia. [10]Infectious Diseases Medical Department, Al Noor Specialist Hospital Makkah, Makkah, MOH, Saudi Arabia. [11]Gastroenterology Department, King Abdul Aziz Hospital Makkah, Makkah, MOH, Saudi Arabia. [12]College of Applied Medical Sciences, Taibah University, Madinah, Saudi Arabia. [13]Department of Medicine, King Abdulaziz University Jeddah, Jeddah, Saudi Arabia. [14]Plan and Research Department, General Directorate of Health Affairs Makkah Region, Makkah, MOH, Saudi Arabia. [15]Vaccines and Immunotherapy Unit, King Fahd Medical Research Center, King Abdulaziz University, Jeddah, Saudi Arabia. [16]Department of Medical Microbiology and Parasitology, Faculty of Medicine, King Abdulaziz University, Jeddah, Saudi Arabia. [17]King Abdullah University of

Science and Technology (KAUST), Computer, Electrical and Mathematical Science and Engineering Division (CEMSE), Thuwal-Jeddah 23955-6900, Saudi Arabia. [18]Center for Genetics and Inherited Diseases, Taibah University, Almadinah Almunwarah, Saudi Arabia. [19]Research Center for Zoonosis Control, Global Institution for Collaborative Research and Education (GI-CoRE), Hokkaido University, N20 W10 Kita-ku, Sapporo 001-0020, Japan. [20]These authors contributed equally: Tobias Mourier, Muhammad Shuaib, Sharif Hala, Sara Mfarrej. ✉email: arnab.pain@kaust.edu.sa

