## [Peer Review File · Nature Communications]

SARS-CoV-2 genomes from Saudi Arabia implicate nucleocapsid mutations in host response and increased viral loadREVIEWER COMMENTS

Reviewer #1 (Remarks to the Author):

Mourier et al. conducted genome monitoring of SARS-CoV-2 in Saudi Arabian by sequencing 892 genomes from March to August 2020, followed by functional analyses of one specific consecutive mutation in N genes.

Major:

1. Given that 7 high-frequency mutations have been found, are other mutations associated with clinical outcomes? Why R203K/G204R was the only one chosen for the follow-up experiments?
2. D614G seems to confound the effect of R203K/G204R and control (Figure3 C), what about other variance? How do you dissect the effect of R203K/G204R from others?
3. Clinical outcomes, including mortality (deceased) and patients admitted to ICU, were used. However, these are largely confounded by the age, treatment or whether the patients with other diseases. The current results of R203K/G204R also can not clearly be associated with mortality after regressing out certain factors. Instead, biochemistry and blood traits are intermediate phenotypes with quantitative values and have been used as clinical diagnosis of the COVID-19 severity. Can the author use these clinical phenotypes to assess the clinical importance and implication of the R203K/G204R and other variants? The authors should be very careful to claim a mutation with "bad" or "good" outcomes, especially with mortality.
4. HEK-293T is kidney-derived cells and only two biological replicates of RNA-sequencing were presented in Figure5. Calu-3 cell line maybe a better choice for the in vitro experiments.
5. Live or reverse engineered virus with R203K/G204R experiment can better verify the effects.

Minor:

1. The SIFT of R203K should also be included in Figure4 A.
2. Why does Figure 4 C only shown the RIP-qPCR of N and E genes?
3. There only one serine phosphorylation sites (S2) outside the LKR region on the N protein, or only S2 was detected by AP-MS in Figure 4G?
4. PCA of figures 13 used all expressed genes or highly variable genes?

Reviewer #2 (Remarks to the Author):

Mourier et al present a wide-ranging investigation into SARS-CoV-2 samples in Saudi Arabia. They use phylogenetic and phylodynamic analyses to investigate population dynamics and importations to the country, with particular attention to the triple SNP R203K/G204R. They go on to show an association of this mutation with severity, mortality and viral load, and, in a biochemical analysis, to investigate possible reasons for this.

Firstly, I have to report that I am not qualified to assess the second, biochemical, half of this manuscript. It is not my field, and I cannot make recommendations about its content. I leave that to other reviewers, and concentrate on the first half, up to the logistic regression analysis.

The paper is ambitious in scope and is very well-written. The treatment of R203K/G204R (up to the logistic regression) is interesting and convincing. I do feel that this manuscript may be trying

to do too much, while still leaving obvious gaps. For example, a GWAS is performed (L218) but is not described in the methods at all. The regression tables for the analyses of associations with severity and mortality (L240) are not given, even in the supplement. The skygrowth model uses travel data as a covariate for no apparent reason (L562), as the fitted values of the coefficients of those are not reported; this is also not described in sufficient methodological detail. The observations surrounding introductions to Saudi Arabia are under-evaluated; they date introductions to SA rather early but do not clearly give statements of uncertainty. At a point in time where the origins of SARS-CoV-2 are being re-investigated, this kind of analysis requires more care than has been put in here. Meanwhile, the skygrowth plot is overinterpreted. My sense is that the authors might do better to concentrate on R203K/G204R from the start, because that seems the strongest strand – but it also involves the half of the paper that I do not have the background to penetrate.

The highest posterior density intervals for the skygrowth demographic reconstruction (figure 1D) are not well-interpreted. The interval for R_{eff} overlaps 1 for almost the entire study period, and thus the statement that it was below 1 from June onwards does not seem entirely justified. The peak in the N_e estimate in late May and early June has a wide interval surrounding it and simulation work (e.g. Hall et al., <https://doi.org/10.1093/ve/vew003>) has shown that dynamics of this sort can be caused simply by stochastic sampling effects. (Indeed, the authors suggest this may simply be an artefact of biased sampling.)

I do not accept the argument that the correlation of samples with the polymorphism to assembled genomes with the SNP is particularly strong evidence against contamination (L229; supplement, L45). Surely if a lab is handling an excess of samples with a given polymorphism, it also increases the probability that where contamination occurs, it will involve that mutation? It may be that the logistical arrangements in hospitals mean this can be ruled out, but this needs to be made explicit if so.

An import from Italy to Saudi Arabia in late January 2020 (L179) is really quite unlikely, unless the authors wish to tap into the current controversy about viral origins. There were very few recorded cases in Italy prior to February and they were all travellers. Indeed, a robust estimate of January would be potentially newsworthy and should not be buried in the text like this. Similarly, the suggestion of an importation from Asia in December 2019 (fig 2C) could be controversial and should not appear as simply a single point on a graph with no uncertainty estimate. I worry that this is simply because of the authors' choice to locate state changes as branch midpoints (L542). I think the authors need to untangle this. On the other hand, I cannot see where the January estimate actually comes from. We are referred to figure S5 but that does not appear to show any transitions to Saudi Arabia at that time.

Minor comments:

L118, fig 1C: The lines for Riyadh and the Eastern Region track each other very very closely. This is counterintuitive. It's also clear that the regions with the most cases provided the fewest samples to this analysis, which is worth some discussion.

L135, fig S3: "The observed numbers of SNPs relative to the Wuhan reference follow the numbers observed in global samples" – I think this is a charitable interpretation of this figure. The line for Saudi sequences is largely flat for months while the global line trends steadily upwards.

L191, fig 2B: What is the y-axis here? Is this a probability density distribution, or a smoothed histogram of counts?

L252: The point that an effect on severity may only lead to an effect on mortality in some clinical circumstances is interesting and perhaps worth stating explicitly.

L536: More details needed on "stratified over time".

L540: It is not at all clear how the 15 trees were produced or what they were used for.

L541: Why use parsimony/treedater here and TreeTime in the next section?

L559: Once again, I do not follow how these 15 phylogenies were generated, what they were used for, or why the number 15 was picked.

Reviewer #3 (Remarks to the Author):

Mourier and colleagues describe the results of SARS-CoV-2 genome sequencing in Saudi Arabia. They present quite good evidence that suggests a nucleocapsid protein mutation is associated with increased viral load and host interactions. While I am not an expert on the biochemical analysis and therefore cannot comment fully on that aspect, the genomic analysis appears to be robust and they present interesting results. I have a few comments that I think will improve the study:

- COVID-19 stands for 'coronavirus disease 19' (i.e. delete the word "infectious").
- I realise the total number of genomes of GISAID gets out of date very quickly but please update.
- The Results refer to Nextstrain clades but it would be helpful to also refer to PANGO equivalents - most researchers are more familiar with PONGA nomenclature.
- Mutations in the nucleocapsid are noted to have a higher incidence in ICU hospitalisations but my understanding is that most of the samples were obtained from hospitals so I am concerned about confounding factors. Nevertheless, the correction with data on patients' ages/co-morbidities is compelling. Were there independent clusters that contained this mutation, i.e. were all cases with this mutation epidemiologically linked? It is interesting that time is a confounder. Can the authors elaborate on this point more?
- Can the authors say how many importations into Saudi Arabia from the genomic data they can detect and how does this compare to other studies? Would be nice to see this in a figure.

Reviewer #4 (Remarks to the Author):

Mourier et al investigate the role of a mutant nucleocapsid protein from SARS-CoV-2 in modulating host interactions and viral load. The authors analyze 892 SARS-CoV-2 genomes from patients in Saudi Arabia from March to August 2020. They further investigate the (R203K/G204R) in the SARS-CoV-2 nucleocapsid (N) protein.

As submitted the proteomics analyses cannot be fully evaluated as data is missing. For all mass spec analyses, the data should be included as searchable files for instance excel sheets. In addition to protein names, peptide counts, peptide/protein scores and quantifications of each replicate need to be included. Results of statistical analysis (ratios, pvalues) should be listed for all proteins and peptides.

How were outliers defined?

The authors cite papers from several other groups to describe their workflow in the method section. To help the reader follow and understand the author's approach to mass spec analysis, data filtering, and data analysis, the authors need to include details on these in addition to the citations.

The raw data needs to be deposited to ProteomeXchange or a similar data repository.

What was the mock control for the AP-MS analysis of mutant and control N protein? A comparison

in table format for proteins identified in the mock and N-proteins samples should be included. What cut-offs did the authors use to remove proteins bound to the mock control to correct for non-specific background binding?

Figure S11 how was the gel stained? This should be indicated in the figure or figure legend.

In Figure 4D, what are the different groups? What type of clustering analysis was performed for Figure 4D?

How was a pvalue calculated for S206 if it was not detected in the N-control (Table S7)?

RESPONSE TO REVIEWER COMMENTS (NCOMMS-21-18178-T)

Our responses are written in blue text.

Reply to Reviewer #1 (Remarks to the Author):

Mourier et al. conducted genome monitoring of SARS-CoV-2 in Saudi Arabian by sequencing 892 genomes from March to August 2020, followed by functional analyses of one specific consecutive mutation in N genes.

We want to thank the reviewer for the constructive comments and suggestions, which turned out to be highly instrumental in improving our manuscript's quality. Please find below our responses to their specific concerns.

Major:

1. Given that 7 high-frequency mutations have been found, are other mutations associated with clinical outcomes? Why R203K/G204R was the only one chosen for the follow-up experiments?

In the initial GWAS analysis (as presented in Figure S5), only the R203K/G204R SNPs showed significant association with mortality in our samples. When testing for associations to mortality, severity, and viral load using regression models, we now include 12 additional SNPs (including all high-frequency SNPs), testing for effects of their own as well as confounding effects on R203K/G204R (Tables S6-S10).

Although other SNPs also to some extent show associations to mortality, severity, and viral load, the R203K/G204R SNPs are consistently found to be significantly associated with these features – except mortality when adjusting for time (which finds no significant association for any tested SNP).

2. D614G seems to confound the effect of R203K/G204R and control (Figure3 C), what about other variance? How do you dissect the effect of R203K/G204R from others?

From our statistical tests (please see above), we can see that the observed association between the R203K/G204R SNPs with severity and viral load is not simply an effect of other SNPs. In the experimental work, the presence of the R203K/G204R SNPs is the only adjustable variable, so any observed cellular or genetic differences are assigned to the effect of the R203K/G204R SNPs.

3. Clinical outcomes, including mortality (deceased) and patients admitted to ICU, were used. However, these are primarily confounded by the age, treatment or whether the patients with other diseases. The current results of R203K/G204R also can not clearly be associated with mortality after regressing out certain factors. Instead, biochemistry and blood traits are intermediate phenotypes with quantitative values and have been used as clinical diagnosis of the COVID-19 severity. Can the author use these clinical phenotypes to assess the clinical importance and implication of the R203K/G204R and other variants? The authors should be very careful to claim a mutation with "bad" or "good" outcomes, especially with mortality.

Unfortunately, we do not have access to such clinical phenotypes. We completely acknowledge that observations such as 'deceased' and 'admission to ICU' are heavily influenced by practical procedures that may differ between hospitals and may further change

over time. The observed statistical associations between R203K/G204R SNPs and 'severity' (as reported by hospitals) should therefore merely be seen as a first, preliminary step in our analysis, followed by the observation of their association with viral load and finally, the experimental investigation of the effect of the SNPs.

4. HEK-293T is kidney-derived cells and only two biological replicates of RNA-sequencing were presented in Figure 5. Calu-3 cell line maybe a better choice for the in vitro experiments.

As suggested by the reviewer, we have now performed the in-vitro transfection and RNA-seq experiment in Calu-3 cell lines. We have used four biological replicates in each condition. The results from this experiment reveal a similar over-expression profile of the immune system and interferon-related genes in the N-mutant condition, as was observed in HEK-293 cell lines. The new data are now included in the main Figure-5 and Figure-S12.

5. Live or reverse engineered virus with R203K/G204R experiment can better verify the effects.

We thank the reviewer for suggesting this important experiment to verify the effects in live or reversed engineered virus systems. First, we would like to inform the reviewer that in our institute we do not have the BSL-3 culture facility, so we could not culture the live virus. Following the reviewer's suggestion, we had established an MTA with the World Reference Center for Emerging Viruses and Arboviruses (WRCEVA) and University of Texas Medical Branch (UTMB) to obtain the plasmids for generating the reverse engineered viruses (as described previously in PMID: 32289263 and PMID: 33514944. This was done in June, 2020 and received the reagents in August. Using these plasmids, we have successfully introduced the desired mutations and generated full length viral RNA for four strains (1. Wildtype (control), 2. R203K/G204R mutant, 3. D614G (as additional control), 4. combination of R203K/G204R mutant and D614G (as control). Now to produce the virus and study the effect we needed to do the next part of the experiment in a BSL-3 facility. For this purpose, we have established a collaboration with Liverpool School of Tropical Medicine (LSTM) in U.K. and the approval paperwork from the UK regulatory bodies is still under process. Second, while we are doing these experiments for the revision and waiting for the regulatory approvals in the UK to handle recombinant SARS-CoV-2 viruses in a LSTM lab, a preprint on *BioRxiv* appeared on October 15, 2021 (<https://www.biorxiv.org/content/10.1101/2021.10.14.464390v1>) reporting the same R203K-G204R mutation and its link to enhancement of replication, fitness, and pathogenesis of SARS-CoV-2, mediated by hyper-phosphorylation of N protein. This study performed the experiment using live virus cell culture and an animal model system. It is important to note that we have obtained the plasmids (for our experiment) from the same group under an MTA and our pre-print has been available in the *MedRxiv* since May, 2021. It is also noteworthy that the above-mentioned paper did not cite our preprint (<https://www.medrxiv.org/content/10.1101/2021.05.06.21256706v1>). Our experiment in BSL-3 facility at LSTM still needs some additional time for execution. However, from the results shared in the preprint (<https://www.biorxiv.org/content/10.1101/2021.10.14.464390v1>), it is clear and reassuring that our novel results are now independently verified by another group in both cell and animal model. Therefore, in this situation we need to publish our paper without further delay as we were the first group to report the functional importance of these N gene mutations (R203K-G204R) in SARS-CoV-2 biology. The results reported in this manuscript are very

similar and overlapping to our work and in fact independently validates the main results that we have reported in our manuscript.

Minor:

1. The SIFT of R203K should also be included in Figure 4 A.

As suggested by the reviewer, we have now included the SIFT values for both R203K and G204R in Figure-4A.

2. Why does Figure 4 C only shown the RIP-qPCR of N and E genes?

The reason for using the N and E gene-specific primers, because these regions are widely used for COVID-19 detection assays, and we had primers available in the lab for those viral sequences. Following the reviewer's concern, we have included additional primers for S and ORF1ab genes in this analysis. Please see the revised Figure-4C.

3. There only one serine phosphorylation sites (S2) outside the LKR region on the N protein, or only S2 was detected by AP-MS in Figure 4G?

There are other serine sites outside the LKR region. The phosphorylation sites detected in our AP-MS include Serine sites (S2, S76, S78, S79, S165, S166, S176, S180, S183, S184, S206) and a Threonine site (T205) (as shown in Table S13).

4. PCA of figure S 13 used all expressed genes or highly variable genes?

The PCA plot in Figure-S13 represents all differentially expressed genes. This plot is updated with new data sets from Calu-3 cells (see the revised version of the PCA plot in Figure-S12).

Reply to Reviewer #2 (Remarks to the Author):

Mourier et al present a wide-ranging investigation into SARS-CoV-2 samples in Saudi Arabia. They use phylogenetic and phylodynamic analyses to investigate population dynamics and importations to the country, with particular attention to the triple SNP R203K/G204R. They go on to show an association of this mutation with severity, mortality and viral load, and, in a biochemical analysis, to investigate possible reasons for this.

Firstly, I have to report that I am not qualified to assess the second, biochemical, half of this manuscript. It is not my field, and I cannot make recommendations about its content. I leave that to other reviewers, and concentrate on the first half, up to the logistic regression analysis.

The paper is ambitious in scope and is very well-written. The treatment of R203K/G204R (up to the logistic regression) is interesting and convincing. I do feel that this manuscript may be trying to do too much while still leaving obvious gaps.

For example, a GWAS is performed (L218) but is not described in the methods at all.

The following has now been added to the methods:

Genome-wide associations tests were carried out as a conventional case-control setup, and for each SNP, a contingency table was constructed as (deceased patients with SNP/ deceased patients without SNP)/(non-deceased patients with SNP/ non-deceased patients without SNP). P-values were calculated using Fisher's exact two-tailed test.

The regression tables for the analyses of associations with severity and mortality (L240) are not given, even in the supplement.

We thank the reviewer for pointing this out. These tables have now been added as supplementary Tables S6-S10.

The skygrowth model uses travel data as a covariate for no apparent reason (L562), as the fitted values of the coefficients of those are not reported; this is also not described in sufficient methodological detail.

The travel data covariate has been removed from the model as it has little effect on the modelled estimates with the changes made to the tree generation for use with skygrowth. The changes made have been described in the revised manuscript (lines 495-530).

The observations surrounding introductions to Saudi Arabia are under-evaluated; they date introductions to S.A. rather early but do not clearly give statements of uncertainty. At a point in time where the origins of SARS-CoV-2 are being re-investigated, this kind of analysis requires more care than has been put in here.

We completely acknowledge the vast uncertainty surrounding the reported introductions. We further realise that any such analysis is strongly affected by and dependent upon the available global samples prior to the origin in Saudi Arabia and will therefore broadly reflect SARS-CoV-2 sampling efforts rather than actual evolutionary events. We have therefore decided to remove this analysis from the manuscript. As a result, panel C has now been removed from Figure 2.

Meanwhile, the skygrowth plot is overinterpreted. My sense is that the authors might do better to concentrate on R203K/G204R from the start, because that seems the strongest strand – but it also involves the half of the paper that I do not have the background to penetrate.

The highest posterior density intervals for the skygrowth demographic reconstruction (figure 1D) are not well-interpreted. The interval for R_{eff} overlaps 1 for almost the entire study period, and thus the statement that it was below 1 from June onwards does not seem entirely justified. The peak in the N_e estimate in late May and early June has a wide interval surrounding it and simulation work (e.g. Hall et al., <https://doi.org/10.1093/ve/vew003>) has shown that dynamics of this sort can be caused simply by stochastic sampling effects. (Indeed, the authors suggest this may simply be an artefact of biased sampling.)

We agree with the general point made by the reviewer and Hall et al. that the median N_e trajectory should not be used to infer a population trend. Consequently, we have both revised the interpretation of the result and provided an alternative justification for our conclusion about the R_{eff} change point, using the point at which greater than 50% of collected traces result in an R estimate below 1 for the first time.

The results by Hall et al., however are not directly applicable to our analysis since they were based on a different population genetic model (skyline as opposed to skygrowth), and the

skygrowth model was designed specifically to estimate changes in growth rates rather than population size and overcome some of the limitations pointed out by Hall *et al.* The caveat about biased sampling is important and biased sampling can confound any phylodynamic analysis, particularly if different sub-populations were sampled at different time points; however, the coalescent method used here should be quite robust to different sampling rates overtime provided the samples were collected randomly.

I do not accept the argument that the correlation of samples with the polymorphism to assembled genomes with the SNP is particularly strong evidence against contamination (L229; supplement, L45). Surely if a lab is handling an excess of samples with a given polymorphism, it also increases the probability that where contamination occurs, it will involve that mutation? It may be that the logistical arrangements in hospitals mean this can be ruled out, but this needs to be made explicit if so.

The idea behind this argument was exactly as pointed out by the reviewer. However, without knowledge about the exact timing and procedures undertaken at the hospitals, we agree that this argument is hard to establish. We have therefore deleted this statement altogether. Instead, we have added the following:

"A fraction of SNPs showing intermediary allele frequencies suggests cases of co-infection with multiple strains (Supplementary Information, Figure S8). Previously, co-infection of SARS-CoV-2 has been reported from observations of bi-allelic haplotypes as well as recombination between genetically distinct lineages²⁶⁻²⁸."

Estimated SNP allele frequencies are now shown in Figure S8.

An import from Italy to Saudi Arabia in late January 2020 (L179) is really quite unlikely unless the authors wish to tap into the current controversy about viral origins. There were very few recorded cases in Italy prior to February, and they were all travellers. Indeed, a robust estimate of January would be potentially newsworthy and should not be buried in the text like this. Similarly, the suggestion of importation from Asia in December 2019 (fig 2C) could be controversial and should not appear as simply a single point on a graph with no uncertainty estimate. I worry that this is simply because of the authors' choice to locate state changes as branch midpoints (L542). I think the authors need to untangle this. On the other hand, I cannot see where the January estimate actually comes from. We are referred to figure S5, but that does not appear to show any transitions to Saudi Arabia at that time.

We completely agree with the issues regarding the geographical origin raised by the reviewer. As stated above, we have removed this analysis altogether from the revised manuscript.

Minor comments:

L118, fig 1C: The lines for Riyadh and the Eastern Region track each other very, very closely. This is counterintuitive. It's also clear that the regions with the most cases provided the fewest samples to this analysis, which is worth some discussion.

The plot in Figure 1C showed stacked values, which was clearly not explained properly. We have changed Figure 1C to a stacked bar chart, and revised the legend accordingly.

L135, fig S3: "The observed numbers of SNPs relative to the Wuhan reference follow the numbers observed in global samples" – I think this is a charitable interpretation of this figure. The line for Saudi sequences is largely flat for months while the global line trends steadily upwards.

This is indeed a fair point. We have revised the sentence to the following:

"The observed numbers of SNPs relative to the Wuhan reference is, in general, lower than the numbers observed in global samples, but except for a period from mid-June to late July, the average number of SNPs in Saudi samples is within one standard deviation of samples deposited in GISAID"

L191, fig 2B: What is the y-axis here? Is this a probability density distribution, or a smoothed histogram of counts?

This is a probability density based on kernel density estimation. Methods for this section and the figure legend have been updated to reflect this more clearly.

L252: The point that an effect on severity may only lead to an effect on mortality in some clinical circumstances is interesting and perhaps worth stating explicitly.

We have added a sentence so that the entire statement reads like this:

"One could speculate if temporal shifts in clinical settings in certain cases have impacted the balance between severity of infection and mortality. Unfortunately, our data set does not allow us to assess if the observed mortality rates are the result of shifts in treatment regimes or admission procedures during the sampling window."

L536: More details needed on "stratified over time".

Further detail on time stratification has been included in the methods:

"Five hundred sequences were selected from available closely related sequences in a time stratified manner, taking every N/500th sequence from the set of N sequences arranged by date, rounded to the nearest integer."

L540: It is not at all clear how the 15 trees were produced or what they were used for.

We have added additional details to the methods:

"These trees were dated using the R package *treedater* (v0.5.0) after collapsing short branch lengths and resolving polytomies randomly fifteen times for each clade with the functions *di2multi* and *multi2di* from the ape R package (v5.5)^{70,71}".

L541: Why use parsimony/*treedater* here and *TreeTime* in the next section?

The second analysis has been removed from the revised manuscript (see above). Hence, this point is no longer applicable.

L559: Once again, I do not follow how these 15 phylogenies were generated, what they were used for, or why the number 15 was picked.

We have updated the skygrowth methodology to reflect uncertainty in the simulated phylogenies more robustly. Multiple steps were taken to ensure that the analysis accounted for uncertainty in phylogeny in terms of topology and dating. Bootstrap was used to estimate a distribution of 1,000 phylogenies, each of which was transformed using the methods described above for fitting the molecular clock. We find results are not sensitive to the number of times polytomies are resolved.

Reply to Reviewer #3 (Remarks to the Author):

Mourier and colleagues describe the results of SARS-CoV-2 genome sequencing in Saudi Arabia. They present quite good evidence that suggests a nucleocapsid protein mutation is associated with increased viral load and host interactions. While I am not an expert on the biochemical analysis and therefore cannot comment fully on that aspect, the genomic analysis appears to be robust and they present interesting results. I have a few comments that I think will improve the study:

- COVID-19 stands for 'coronavirus disease 19' (i.e. delete the word "infectious").

This has been deleted.

- I realise the total number of genomes of GISAID gets out of date very quickly but please update.

This has been updated before submission of the revised manuscript.

- The Results refer to Nextstrain clades, but it would be helpful to also refer to PANGO equivalents - most researchers are more familiar with PONGA nomenclature.

Although this is of course not a 1:1 relationship, we have now added an approximate conversion to Pangolin lineages as Table S5

- Mutations in the nucleocapsid are noted to have a higher incidence in ICU hospitalisations but my understanding is that most of the samples were obtained from hospitals so I am concerned about confounding factors. Nevertheless, the correction with data on patients' ages/co-morbidities is compelling. Were there independent clusters that contained this mutation, i.e. were all cases with this mutation epidemiologically linked? It is interesting that time is a confounder. Can the authors elaborate on this point more?

Unfortunately, the available data does not allow us to assess the specific epidemiology of the hospital cases. Similarly, although we can see that time is a confounding factor we are unable to address the exact details of this. We have extended the manuscript slightly to contain the following (see also response to reviewer #2): "One could speculate if temporal shifts in clinical settings in certain cases have impacted the balance between severity of infection and mortality. Unfortunately, our data set does not allow us to assess if the observed mortality rates are the result of shifts in treatment regimens or admission procedures during the sampling window."

- Can the authors say how many importations into Saudi Arabia from the genomic data they can detect and how does this compare to other studies? Would be nice to see this in a figure.

The uncertainty surrounding the analysis of importation events has prompted us to remove the analysis of specific events (please see response to reviewer #2 above). To our knowledge, no other studies have addressed this.

Reply to Reviewer #4 (Remarks to the Author):

Mourier et al investigate the role of a mutant nucleocapsid protein from SARS-CoV-2 in modulating host interactions and viral load. The authors analyse 892 SARS-CoV-2 genomes from patients in Saudi Arabia from March to August 2020. They further investigate the (R203K/G204R) in the SARS-CoV-2 nucleocapsid (N) protein.

First, we thank the reviewer for this thorough review, constructive comments, and suggestions on our manuscript, which allowed us to improve the quality of the revised manuscript. Please find below our responses to specific concerns.

As submitted the proteomics analyses cannot be fully evaluated as data is missing. For all mass spec analyses, the data should be included as searchable files for instance excel sheets. In addition to protein names, peptide counts, peptide/protein scores and quantifications of each replicate need to be included. Results of statistical analysis (ratios, p-values) should be listed for all proteins and peptides.

As suggested by the reviewer, we have now included all the mass spec data (raw) in excel sheets (please see Table S11 for AP-MS data of N-protein and Mock-control). This table shows all the requested information (such as protein names, peptide counts, peptide/protein scores, quantifications of each replicate, and statistics). After background correction (by removing proteins identified in mock-control), the differential interactome of mutant and control N-protein is shown in Table S12 (this table shows all data points, including significantly changed and non-significant changed interacting partners with p-values and adjusted p-values information).

Also, all the affinity-purification mass spectrometry (AP-MS) proteomics data have been deposited to the ProteomeXchange Consortium via the PRIDE partner repository with the dataset identifier PXD027168. The reviewers can access the data with the following credential.

Reviewer account details:

Username: reviewer_pxd027168@ebi.ac.uk

Password: CsX6Romi

For further clarification, the sample file naming format and the datasets associated with the MaxQuant analysis are represented in Table S17.

How were outliers defined?

For the affinity-purification mass spectrometry (AP-MS) differential interaction analysis we have used three biological replicates in each condition (1. N protein wildtype AP-MS, 2. N protein KR-mutant AP-MS, and 3. Mock AP-MS). All these three replicates were included in the final analysis, so no outliers were removed from these data sets. Similarly, for the detection of protein phosphorylation, we used five biological replicates (with three technical

replicates in each condition). Please see Table S17 for the sample naming information that are deposited to ProteomeXchange (identifier: PXD027168).

The authors cite papers from several other groups to describe their workflow in the method section. To help the reader follow and understand the author's approach to mass spec analysis, data filtering, and data analysis, the authors need to include details on these in addition to the citations.

We thank the reviewer for this important suggestion. As suggested by the reviewer, we have now included detail in the method section of the revised manuscript (please see Line 596-617).

The raw data needs to be deposited to ProteomeXchange or a similar data repository.

As mentioned before, we have now deposited all the raw mass spectrometry data sets to the ProteomeXchange Consortium via the PRIDE partner repository with the dataset identifier PXD027168. The reviewers can access the data with the following credential.

Reviewer account details:

Username: reviewer_pxd027168@ebi.ac.uk

Password: CsX6Romi

What was the mock control for the AP-MS analysis of mutant and control N protein?

For the mock control for the affinity mass spectrometry (AP-MS), cells were transfected with the same plasmid vector without the N gene (mutant and control). Similar to N-(mutant & control) AP-MS, we used three biological replicates for the mock experiment.

A comparison in table format for proteins identified in the mock and N-proteins samples should be included.

As suggested, we have included a comparison in table format for proteins identified in the mock and N-proteins samples (Please see Table S18 for this comparison).

What cut-offs did the authors use to remove proteins bound to the mock control to correct for non-specific background binding?

To correct for non-specific background binding, any protein detected in the mock-control experiment were removed from differential interaction analysis (Please see Table S18).

Figure S11 how was the gel stained? This should be indicated in the figure or figure legend.

We have done silver staining for the gel in Figure-S11. We have now included this information in both the figure and legend (please see the revised figure as Figure-S10).

In Figure 4D, what are the different groups? What type of clustering analysis was performed for Figure 4D?

The clustering in Figure-4D of the first version of the manuscript represented only the group based on their expression (in this case, interaction scores), and they were generated automatically by the tool (LFQ-Analyst) used for differential interaction analysis. In the revised manuscript, we have removed this clustering information (please see the updated revised Figure-4D).

How was a pvalue calculated for S206 if it was not detected in the N-control (Table S7)?

This was reported by error while we calculated the non-significant change in the other phosphorylation sites. In the revised manuscript, we have only shown whether phosphorylation was detected or not at each site. Please see the updated Figure-4G and Table-S13 (complete raw phospho(STY) sites data are available with the identifier PXD027168).

REVIEWERS' COMMENTS

Reviewer #1 (Remarks to the Author):

The authors have replied to my comments and some have been addressed and the others were reported elsewhere, indicating the biological significance of these loci.

Reviewer #2 (Remarks to the Author):

I thank the authors for the work done on the revisions here; the manuscript is much improved in this version. I have some follow-up comments and suggestions:

L173: It would be possible to do some kind of probabilistic maximum likelihood estimation (e.g. Markov Jumps) to quantify the level of uncertainty in the estimated number of times that the mutation occurred.

L178: The patterns in the global data (peak in July, then decline to November followed by another increase) are quite distinct from those in SA (peak in May, then decline to apparent extinction), despite what the text argues. Discrepancies like this tend to beg questions in readers' minds.

L193: The new version of the text regarding possible contamination seems to come out of nowhere and it is quite unclear what point it is making, its connection to the preceding sentence, or what purpose it even serves. It should be also made clear that while within-sample MAFs close to 0 or 1 likely indicate contamination, those that are literally 0 or 1 do not (figure S8, legend).

L350: Discuss this point a bit. Why would the severity effect persist when controlling for time period, but the mortality effect disappear?

Figures: The appearance in legends of the text "Wuhan reference" to represent sequences without R203K/G204R is quite confusing whenever it appears, and I suggest choosing something else.

Reviewer #4 (Remarks to the Author):

The authors have addressed all of my concerns.

RESPONSE TO REVIEWER COMMENTS

Reply to Reviewer #2 (Remarks to the Author):

I thank the authors for the work done on the revisions here; the manuscript is much improved in this version. I have some follow-up comments and suggestions:

L173: It would be possible to do some kind of probabilistic maximum likelihood estimation (e.g. Markov Jumps) to quantify the level of uncertainty in the estimated number of times that the mutation occurred.

We thank the reviewer for this important suggestion. While we agree that such an approach would be interesting, we believe that this would constitute almost an entire study in itself and is beyond the scope of the present study.

L178: The patterns in the global data (peak in July, then decline to November followed by another increase) are quite distinct from those in SA (peak in May, then decline to apparent extinction), despite what the text argues. Discrepancies like this tend to beg questions in readers' minds.

We agree with the reviewer's concern. As suggested, this is now revised as below:

"Within our sampling window we observe an apparent transient increase in the frequency of R203K/G204R SNPs (Fig. 3a) in accordance with earlier observations^{18,25}. In the global data, the peak in R203K/G204R frequency is slightly delayed compared to samples from Saudi Arabia. The initial global peak is observed in July 2020 followed by a decline until the fall of 2020, where the R203K/G204R SNPs once again increased along with the Spike protein Y501N mutation in the B1.1.17 lineage 17 (Fig. 3a)."

L193: The new version of the text regarding possible contamination seems to come out of nowhere and it is quite unclear what point it is making, its connection to the preceding sentence, or what purpose it even serves. It should be also made clear that while within-sample MAFs close to 0 or 1 likely indicate contamination, those that are literally 0 or 1 do not (figure S8, legend).

We absolutely agree with this. Having initially moved this section to another place in the manuscript, we now realise that indeed it does not fit well with the rest of the manuscript. And as the reviewer notes, the point of it is not clear. We have therefore decided to remove this section (along with Supplementary Fig. S8) from the manuscript.

L350: Discuss this point a bit. Why would the severity effect persist when controlling for time period, but the mortality effect disappear?

We have added the following to the discussion:

"The association of N protein R203K and G204R mutations with higher viral load persists after adjusting for time but not with mortality, suggesting that the mortality rate of severe infections may be influenced by other factors such as changes in treatment regimes as well as the complexities of host response to SARS-CoV-2 infection. Unfortunately, the available data do not allow a further assessment of this."

Figures: The appearance in legends of the text "Wuhan reference" to represent sequences without R203K/G204R is quite confusing whenever it appears, and I suggest choosing something else.

In figure 3, we have changed this to "Wuhan-Hu-1 isolate reference" in panel c, and simply to "reference" in panel d.
Throughout the main text, "Wuhan" is now consistently written as "Wuhan-Hu-1"